# On the Global Convergence of Gradient Descent for Over-parameterized Models using Optimal Transport

**Lénaïc Chizat**
INRIA, ENS, PSL Research University
Paris, France
lenaic.chizat@inria.fr

**Francis Bach**
INRIA, ENS, PSL Research University
Paris, France
francis.bach@inria.fr

## Abstract

Many tasks in machine learning and signal processing can be solved by minimizing a convex function of a measure. This includes sparse spikes deconvolution or training a neural network with a single hidden layer. For these problems, we study a simple minimization method: the unknown measure is discretized into a mixture of particles and a continuous-time gradient descent is performed on their weights and positions. This is an idealization of the usual way to train neural networks with a large hidden layer. We show that, when initialized correctly and in the many-particle limit, this gradient flow, although non-convex, converges to global minimizers. The proof involves Wasserstein gradient flows, a by-product of optimal transport theory. Numerical experiments show that this asymptotic behavior is already at play for a reasonable number of particles, even in high dimension.

## 1 Introduction

A classical task in machine learning and signal processing is to search for an element in a Hilbert space $\mathcal{F}$ that minimizes a smooth, convex *loss* function $R : \mathcal{F} \to \mathbb{R}_+$ and that is a linear combination of a few elements from a large given parameterized set $\{\phi(\theta)\}_{\theta \in \Theta} \subset \mathcal{F}$. A general formulation of this problem is to describe the linear combination through an unknown signed measure $\mu$ on the parameter space and to solve for

$$J^* = \min_{\mu \in \mathcal{M}(\Theta)} J(\mu), \qquad J(\mu) := R\left(\int \phi \mathrm{d}\mu\right) + G(\mu) \qquad (1)$$

where $\mathcal{M}(\Theta)$ is the set of signed measures on the parameter space $\Theta$ and $G : \mathcal{M}(\Theta) \to \mathbb{R}$ is an optional convex regularizer, typically the total variation norm when sparse solutions are preferred. In this paper, we consider the *infinite-dimensional* case where the parameter space $\Theta$ is a domain of $\mathbb{R}^d$ and $\theta \mapsto \phi(\theta)$ is differentiable. This framework covers:

- Training *neural networks with a single hidden layer*, where the goal is to select, within a specific class, a function that maps features in $\mathbb{R}^{d-1}$ to labels in $\mathbb{R}$, from the observation of a joint distribution of features and labels. This corresponds to $\mathcal{F}$ being the space of square-integrable real-valued functions on $\mathbb{R}^{d-1}$, $R$ being, e.g., the quadratic or the logistic loss function, and $\phi(\theta) : x \mapsto \sigma(\sum_{i=1}^{d-1} \theta_i x_i + \theta_d)$, with an activation function $\sigma : \mathbb{R} \to \mathbb{R}$. Common choices are the sigmoid function or the rectified linear unit [18, 14], see more details in Section 4.2.

- *Sparse spikes deconvolution*, where one attempts to recover a signal which is a mixture of impulses on $\Theta$ given a noisy and filtered observation $y$ (a square-integrable function on $\Theta$). This corresponds to $\mathcal{F}$ being the space of square-integrable real-valued functions on $\mathbb{R}^d$, defining $\phi(\theta) : x \mapsto \psi(x - \theta)$ the translations of the filter impulse response $\psi$

and $R(f) = (1/2\lambda)\|f - y\|_{L^2}^2$, for some $\lambda > 0$ that depends on the estimated noise level. Solving (1) allows then to reconstruct the mixture of impulses with some guarantees [12, 13].

- Low-rank tensor decomposition [16], recovering mixture models from sketches [26], see [6] for a detailed list of other applications. For example, with symmetric matrices, $\mathcal{F} = \mathbb{R}^{d \times d}$ and $\Phi(\theta) = \theta\theta^\top$, we recover low-rank matrix decompositions [15].

## 1.1 Review of optimization methods and previous work

While (1) is a convex problem, finding approximate minimizers is hard as the variable is infinite-dimensional. Several lines of work provide optimization methods but with strong limitations.

**Conditional gradient / Frank-Wolfe.** This approach tackles a variant of (1) where the regularization term is replaced by an upper bound on the total variation norm; the associated constraint set is the convex hull of all Diracs and negatives of Diracs at elements of $\theta \in \Theta$, and thus adapted to conditional gradient algorithms [19]. At each iteration, one adds a new particle by solving a linear minimization problem over the constraint set (which correspond to finding a particle $\theta \in \Theta$), and then updates the weights. The resulting iterates are sparse and there is a guaranteed sublinear convergence rate of the objective function to its minimum. However, the linear minimization subroutine is hard to perform in general : it is for instance NP-hard for neural networks with homogeneous activations [4]. One thus generally resorts to space gridding (in low dimension) or to approximate steps, akin to boosting [36]. The practical behavior is improved with nonconvex updates [6, 7] reminiscent of the flow studied below.

**Semidefinite hierarchy.** Another approach is to parameterize the unknown measure by its sequence of moments. The space of such sequences is characterized by a hierarchy of SDP-representable necessary conditions. This approach concerns a large class of *generalized moment problems* [22] and can be adapted to deal with special instances of (1) [9]. It is however restricted to $\phi$ which are combinations of few polynomial moments, and its complexity explodes exponentially with the dimension $d$. For $d \geq 2$, convergence to a global minimizer is only guaranteed asymptotically, similarly to the results of the present paper.

**Particle gradient descent.** A third approach, which exploits the differentiability of $\phi$, consists in discretizing the unknown measure $\mu$ as a mixture of $m$ particles parameterized by their positions and weights. This corresponds to the finite-dimensional problem

$$\min_{\substack{\boldsymbol{w} \in \mathbb{R}^m \\ \boldsymbol{\theta} \in \Theta^m}} J_m(\boldsymbol{w}, \boldsymbol{\theta}) \qquad \text{where} \qquad J_m(\boldsymbol{w}, \boldsymbol{\theta}) := J\left(\frac{1}{m} \sum_{i=1}^m w_i \delta_{\theta_i}\right), \qquad (2)$$

which can then be solved by classical gradient descent-based algorithms. This method is simple to implement and is widely used for the task of neural network training but, a priori, we may only hope to converge to local minima since $J_m$ is non-convex. *Our goal is to show that this method also benefits from the convex structure of* (1) *and enjoys an asymptotical global optimality guarantee.*

There is a recent literature on global optimality results for (2) in the specific task of training neural networks. It is known that in this context, $J_m$ has less, or no, local minima in an over-parameterization regime and stochastic gradient descent (SGD) finds a global minimizer under restrictive assumptions [34, 35, 33, 23]; see [33] for an account of recent results. Our approach is not directly comparable to these works: it is more abstract and nonquantitative—we study an ideal dynamics that one can only hope to approximate—but also much more generic. Our objective, in the space of measures, *has* many local minima, but we build gradient flows that avoids them, relying mainly on the homogeneity properties of $J_m$ (see [16, 20] for other uses of homogeneity in non-convex optimization). The novelty is to see (2) as a discretization of (1)—a point of view also present in [25] but not yet exploited for global optimality guarantees.

## 1.2 Organization of the paper and summary of contributions

Our goal is to explain when and why the non-convex particle gradient descent finds global minima. We do so by studying the many-particle limit $m \to \infty$ of the gradient flow of $J_m$. More specifically:

- In Section 2, we introduce a more general class of problems and study the many-particle limit of the associated particle gradient flow. This limit is characterized as a *Wasserstein gradient flow* (Theorem 2.6), an object which is a by-product of optimal transport theory.

- In Section 3, under assumptions on $\phi$ and the initialization, we prove that if this Wasserstein gradient flow converges, then the limit is a global minimizer of $J$. Under the same conditions, it follows that if $(\boldsymbol{w}^{(m)}(t), \boldsymbol{\theta}^{(m)}(t))_{t \geq 0}$ are gradient flows for $J_m$ suitably initialized, then

$$\lim_{m,t \to \infty} J(\mu_{m,t}) = J^* \qquad \text{where} \qquad \mu_{m,t} = \frac{1}{m} \sum_{i=1}^{m} w_i^{(m)}(t) \delta_{\theta_i^{(m)}(t)}.$$

- Two different settings that leverage the structure of $\phi$ are treated: the 2-*homogeneous* and the *partially* 1-*homogeneous* case. In Section 4, we apply these results to sparse deconvolution and training neural networks with a single hidden layer, with sigmoid or ReLU activation function. In each case, our result prescribes conditions on the initialization pattern.

- We perform simple numerical experiments that indicate that this asymptotic regime is already at play for small values of $m$, even for high-dimensional problems. The method behaves incomparably better than simply optimizing on the weights with a very large set of fixed particles.

Our focus on qualitative results might be surprising for an optimization paper, but we believe that this is an insightful first step given the hardness and the generality of the problem. We suggest to understand our result as a first *consistency principle* for practical and a commonly used non-convex optimization methods. While we focus on the idealistic setting of a *continuous-time* gradient flow with *exact* gradients, this is expected to reflect the behavior of first order descent algorithms, as they are known to approximate the former: see [31] for (accelerated) gradient descent and [21, Thm. 2.1] for SGD.

**Notation.** Scalar products and norms are denoted by $\cdot$ and $|\cdot|$ respectively in $\mathbb{R}^d$, and by $\langle \cdot, \cdot \rangle$ and $\|\cdot\|$ in the Hilbert space $\mathcal{F}$. Norms of linear operators are also denoted by $\|\cdot\|$. The differential of a function $f$ at a point $x$ is denoted $df_x$. We write $\mathcal{M}(\mathbb{R}^d)$ for the set of finite signed Borel measures on $\mathbb{R}^d$, $\delta_x$ is a Dirac mass at a point $x$ and $\mathcal{P}_2(\mathbb{R}^d)$ is the set of probability measures endowed with the Wasserstein distance $W_2$ (see Appendix A).

**Recent related work.** Several independent works [24, 28, 32] have studied the many-particle limit of training a neural network with a single large hidden layer and a quadratic loss $R$. Their main focus is on quantifying the convergence of SGD or noisy SGD to the limit trajectory, which is precisely a mean-field limit in this case. Since in our approach this limit is mostly an intermediate step necessary to state our global convergence theorems, it is not studied extensively for itself. These papers thus provide a solid complement to Section 2.4 (a difference is that we do not assume that $R$ is quadratic nor that $V$ is differentiable). Also, [24] proves a quantitative global convergence result for noisy SGD to an approximate minimizer: we stress that our results are of a different nature, as they rely on homogeneity and not on the mixing effect of noise.

## 2 Particle gradient flows and many-particle limit

### 2.1 Main problem and assumptions

From now on, we consider the following class of problems on the space of *non-negative* finite measures on a domain $\Omega \subset \mathbb{R}^d$ which, as explained below, is more general than (1):

$$F^* = \min_{\mu \in \mathcal{M}_+(\Omega)} F(\mu) \qquad \text{where} \qquad F(\mu) = R\left(\int \Phi \mathrm{d}\mu\right) + \int V \mathrm{d}\mu, \tag{3}$$

and we make the following assumptions.

**Assumptions 2.1.** *$\mathcal{F}$ is a separable Hilbert space, $\Omega \subset \mathbb{R}^d$ is the closure of a convex open set, and*

*(i) (smooth loss) $R : \mathcal{F} \to \mathbb{R}_+$ is differentiable, with a differential $dR$ that is Lipschitz on bounded sets and bounded on sublevel sets,*

*(ii)* (basic regularity) $\Phi : \Omega \to \mathcal{F}$ is (Fréchet) differentiable, $V : \Omega \to \mathbb{R}_+$ is semiconvex[1], and

*(iii)* (locally Lipschitz derivatives with sublinear growth) *there exists a family $(Q_r)_{r>0}$ of nested nonempty closed convex subsets of $\Omega$ such that:*

  *(a) $\{u \in \Omega \;;\; \mathrm{dist}(u, Q_r) \leq r'\} \subset Q_{r+r'}$ for all $r, r' > 0$,*

  *(b) $\Phi$ and $V$ are bounded and $d\Phi$ is Lipschitz on each $Q_r$, and*

  *(c) there exists $C_1, C_2 > 0$ such that $\sup_{u \in Q_r}(\|d\Phi_u\| + \|\partial V(u)\|) \leq C_1 + C_2 r$ for all $r > 0$, where $\|\partial V(u)\|$ stands for the maximal norm of an element in $\partial V(u)$.*

Assumption 2.1-(iii) reduces to classical local Lipschitzness and growth assumptions on $d\Phi$ and $\partial V$ if the nested sets $(Q_r)_r$ are the balls of radius $r$, but unbounded sets $Q_r$ are also allowed. These sets are a technical tool used later to confine the gradient flows in areas where gradients are well-controlled. By convention, we set $F(\mu) = \infty$ if $\mu$ is not concentrated on $\Omega$. Also, the integral $\int \Phi d\mu$ is a *Bochner integral* [10, App. E6]. It yields a well-defined value in $\mathcal{F}$ whenever $\Phi$ is measurable and $\int \|\phi\| d|\mu| < \infty$. Otherwise, we also set $F(\mu) = \infty$ by convention.

**Recovering** (1) **through lifting.** It is shown in Appendix A.2 that, for a class of admissible regularizers $G$ containing the total variation norm, problem (1) admits an equivalent formulation as (3). Indeed, consider the *lifted* domain $\Omega = \mathbb{R} \times \Theta$, the function $\Phi(w, \theta) = w\phi(\theta)$ and $V(w, \theta) = |w|$. Then $J^*$ equals $F^*$ and given a minimizer of one of the problems, one can easily build minimizers for the other. This equivalent *lifted* formulation removes the asymmetry between weight and position— weight becomes just another coordinate of a particle's position. This is the right point of view for our purpose and this is why $F$ is our central object of study in the following.

**Homogeneity.** The functions $\Phi$ and $V$ obtained through the lifting share the property of being positively 1-homogeneous in the variable $w$. A function $f$ between vector spaces is said positively $p$-homogeneous when for all $\lambda > 0$ and argument $x$, it holds $f(\lambda x) = \lambda^p f(x)$. This property is central for our global convergence results (but is not needed throughout Section 2).

## 2.2 Particle gradient flow

We first consider an initial measure which is a mixture of particles—an atomic measure— and define the initial object in our construction: the *particle gradient flow*. For a number $m \in \mathbb{N}$ of particles, and a vector $\mathbf{u} \in \Omega^m$ of positions, this is the gradient flow of

$$F_m(\mathbf{u}) := F\left(\frac{1}{m}\sum_{i=1}^m \delta_{\mathbf{u}_i}\right) = R\left(\frac{1}{m}\sum_{i=1}^m \Phi(\mathbf{u}_i)\right) + \frac{1}{m}\sum_{i=1}^m V(\mathbf{u}_i), \tag{4}$$

or, more precisely, its *subgradient flow* because $V$ can be non-smooth. We recall that a *subgradient* of a (possibly non-convex) function $f : \mathbb{R}^d \to \bar{\mathbb{R}}$ at a point $u_0 \in \mathbb{R}^d$ is a $p \in \mathbb{R}^d$ satisfying $f(u) \geq f(u_0) + p \cdot (u - u_0) + o(u - u_0)$ for all $u \in \mathbb{R}^d$. The set of subgradients at $u$ is a closed convex set called the *subdifferential* of $f$ at $u$ denoted $\partial f(u)$ [27].

**Definition 2.2** (Particle gradient flow). *A gradient flow for the functional $F_m$ is an absolutely continuous[2] path $\mathbf{u} : \mathbb{R}_+ \to \Omega^m$ which satisfies $\mathbf{u}'(t) \in -m \, \partial F_m(\mathbf{u}(t))$ for almost every $t \geq 0$.*

This definition uses a subgradient scaled by $m$, which is the subgradient relative to the scalar product on $(\mathbb{R}^d)^m$ scaled by $1/m$: this normalization amounts to assigning a mass $1/m$ to each particle and is convenient for taking the many-particle limit $m \to \infty$. We now state basic properties of this object.

**Proposition 2.3.** *For any initialization $\mathbf{u}(0) \in \Omega^m$, there exists a unique gradient flow $\mathbf{u} : \mathbb{R}_+ \to \Omega^m$ for $F_m$. Moreover, for almost every $t > 0$, it holds $\frac{d}{ds}F_m(\mathbf{u}(s))|_{s=t} = -|\mathbf{u}'(t)|^2$ and the velocity of the $i$-th particle is given by $\mathbf{u}'_i(t) = v_t(\mathbf{u}_i(t))$, where for $u \in \Omega$ and $\mu_{m,t} := (1/m)\sum_{i=1}^m \delta_{\mathbf{u}_i(t)}$,*

$$v_t(u) = \tilde{v}_t(u) - \mathrm{proj}_{\partial V(u)}(\tilde{v}_t(u)) \quad \text{with} \quad \tilde{v}_t(u) = -\left[\left\langle R'\left(\int \Phi d\mu_{m,t}\right), \partial_j \Phi(u)\right\rangle\right]_{j=1}^d. \tag{5}$$

The expression of the velocity involves a projection because gradient flows select subgradients of minimal norm [29]. We have denoted by $R'(f) \in \mathcal{F}$ the gradient of $R$ at $f \in \mathcal{F}$ and by $\partial_j \Phi(u) \in \mathcal{F}$ the differential $d\Phi_u$ applied to the $j$-th vector of the canonical basis of $\mathbb{R}^d$. Note that $[\tilde{v}_t(\mathbf{u}_i)]_{i=1}^m$ is (minus) the gradient of the first term in (4) : when $V$ is differentiable, we have $v_t(u) = \tilde{v}_t(u) - \nabla V(u)$ and we recover the classical gradient of (4). When $V$ is non-smooth, this gradient flow can be understood as a continuous-time version of the forward-backward minimization algorithm [11].

## 2.3 Wasserstein gradient flow

The fact that the velocity of each particle can be expressed as the evaluation of a velocity field (Eq. (5)) makes it easy, at least formally, to generalize the particle gradient flow to arbitrary measure-valued initializations—not just atomic ones. On the one hand, the evolution of a time-dependent measure $(\mu_t)_t$ under the action of instantaneous velocity fields $(v_t)_{t \geq 0}$ can be formalized by a conservation of mass equation, known as the *continuity equation*, that reads $\partial_t \mu_t = -\mathrm{div}(v_t \mu_t)$ where $\mathrm{div}$ is the divergence operator[3] (see Appendix B). On the other hand, there is a direct link between the velocity field (5) and the functional $F$. The differential of $F$ evaluated at $\mu \in \mathcal{M}(\Omega)$ is represented by the function $F'(\mu) : \Omega \to \mathbb{R}$ defined as

$$F'(\mu)(u) := \left\langle R'\left(\int \Phi \mathrm{d}\mu\right), \Phi(u) \right\rangle + V(u).$$

Thus $v_t$ is simply a field of (minus) subgradients of $F'(\mu_{m,t})$—it is in fact the field of minimal norm subgradients. We write this relation $v_t \in -\partial F'(\mu_{m,t})$. The set $\partial F'$ is called the *Wasserstein subdifferential* of $F$, as it can be interpreted as the subdifferential of $F$ relatively to the Wasserstein metric on $\mathcal{P}_2(\Omega)$ (see Appendix B.2.1). We thus expect that for initializations with arbitrary probability distributions, the generalization of the gradient flow coincides with the following object.

**Definition 2.4** (Wasserstein gradient flow). *A Wasserstein gradient flow for the functional $F$ on a time interval $[0, T[$ is an absolutely continuous path $(\mu_t)_{t \in [0,T[}$ in $\mathcal{P}_2(\Omega)$ that satisfies, distributionally on $[0, T[ \times \Omega^d$,*

$$\partial_t \mu_t = -\mathrm{div}(v_t \mu_t) \quad where \quad v_t \in -\partial F'(\mu_t). \tag{6}$$

This is a proper generalization of Definition 2.2 since, whenever $(\mathbf{u}(t))_{t \geq 0}$ is a particle gradient flow for $F_m$, then $t \mapsto \mu_{m,t} := \frac{1}{m} \sum_{i=1}^m \delta_{\mathbf{u}_i(t)}$ is a Wasserstein gradient flow for $F$ in the sense of Definition 2.4 (see Proposition B.1). By leveraging the abstract theory of gradient flows developed in [3], we show in Appendix B.2.1 that these Wasserstein gradient flows are well-defined.

**Proposition 2.5** (Existence and uniqueness). *Under Assumptions 2.1, if $\mu_0 \in \mathcal{P}_2(\Omega)$ is concentrated on a set $Q_{r_0} \subset \Omega$, then there exists a unique Wasserstein gradient flow $(\mu_t)_{t \geq 0}$ for $F$ starting from $\mu_0$. It satisfies the continuity equation with the velocity field defined in (5) (with $\mu_t$ in place of $\mu_{m,t}$).*

Note that the condition on the initialization is automatically satisfied in Proposition 2.3 because there the initial measure has a finite discrete support: it is thus contained in any $Q_r$ for $r > 0$ large enough.

## 2.4 Many-particle limit

We now characterize the many-particle limit of classical gradient flows, under Assumptions 2.1.

**Theorem 2.6** (Many-particle limit). *Consider $(t \mapsto \mathbf{u}_m(t))_{m \in \mathbb{N}}$ a sequence of classical gradient flows for $F_m$ initialized in a set $Q_{r_0} \subset \Omega$. If $\mu_{m,0}$ converges to some $\mu_0 \in \mathcal{P}_2(\Omega)$ for the Wasserstein distance $W_2$, then $(\mu_{m,t})_t$ converges, as $m \to \infty$, to the unique Wasserstein gradient flow of $F$ starting from $\mu_0$.*

Given a measure $\mu_0 \in \mathcal{P}_2(Q_{r_0})$, an example for the sequence $\mathbf{u}_m(0)$ is $\mathbf{u}_m(0) = (u_1, \ldots, u_m)$ where $u_1, u_2, \ldots, u_m$ are independent samples distributed according to $\mu_0$. By the law of large numbers for empirical distributions, the sequence of empirical distributions $\mu_{m,0} = \frac{1}{m} \sum_{i=1}^m \delta_{u_i}$ converges (almost surely, for $W_2$) to $\mu_0$. In particular, our proof of Theorem 2.6 gives an alternative proof of the existence claim in Proposition 2.5 (the latter remains necessary for the uniqueness of the limit).

# 3 Convergence to global minimizers

## 3.1 General idea

As can be seen from Definition 2.4, a probability measure $\mu \in \mathcal{P}_2(\Omega)$ is a stationary point of a Wasserstein gradient flow if and only if $0 \in \partial F'(\mu)(u)$ for $\mu$-a.e. $u \in \Omega$. It is proved in [25] that these stationary points are, in some cases, optimal over probabilities that have a smaller support. However, they are not in general global minimizers of $F$ over $\mathcal{M}_+(\Omega)$, even when $R$ is convex. Such global minimizers are indeed characterized as follows.

**Proposition 3.1** (Minimizers). *Assume that $R$ is convex. A measure $\mu \in \mathcal{M}_+(\Omega)$ such that $F(\mu) < \infty$ minimizes $F$ on $\mathcal{M}_+(\Omega)$ iff $F'(\mu) \geq 0$ and $F'(\mu)(u) = 0$ for $\mu$-a.e. $u \in \Omega$.*

Despite these strong differences between stationarity and global optimality, we show in this section that Wasserstein gradient flows converge to global minimizers, under two main conditions:

- *On the structure*: $\Phi$ and $V$ must share a homogeneity direction (see Section 2.1 for the definition of homogeneity), and

- *On the initialization*: the support of the initialization of the Wasserstein gradient flow satisfies a "separation" property. This property is preserved throughout the dynamic and, combined with homogeneity, allows to escape from neighborhoods of non-optimal points.

We turn these general ideas into concrete statements for two cases of interest, that exhibit different structures and behaviors: (i) when $\Phi$ and $V$ are positively 2-homogeneous and (ii) when $\Phi$ and $V$ are positively 1-homogeneous with respect to one variable.

## 3.2 The 2-homogeneous case

In the 2-homogeneous case a rich structure emerges, where the $(d-1)$-dimensional sphere $\mathbb{S}^{d-1} \subset \mathbb{R}^d$ plays a special role. This covers the case of lifted problems of Section 2.1 when $\phi$ is 1-homogeneous and neural networks with ReLU activation functions.

**Assumptions 3.2.** *The domain is $\Omega = \mathbb{R}^d$ with $d \geq 2$ and $\Phi$ is differentiable with $d\Phi$ locally Lipschitz, $V$ is semiconvex and $V$ and $\Phi$ are both positively 2-homogeneous. Moreover,*

*(i) (smooth convex loss) The loss $R$ is convex, differentiable with differential $dR$ Lipschitz on bounded sets and bounded on sublevel sets,*

*(ii) (Sard-type regularity) For all $f \in \mathcal{F}$, the set of regular values[4] of $\theta \in \mathbb{S}^{d-1} \mapsto \langle f, \Phi(\theta) \rangle + V(\theta)$ is dense in its range (it is in fact sufficient that this holds for functions $f$ which are of the form $f = R'(\int \Phi d\mu)$ for some $\mu \in \mathcal{M}_+(\Omega)$).*

Taking the balls of radius $r > 0$ as the family $(Q_r)_{r>0}$, these assumptions imply Assumptions 2.1. We believe that Assumption 3.2-(ii) is not of practical importance: it is only used to avoid some pathological cases in the proof of Theorem 3.3. By applying Morse-Sard's lemma [1], it is anyways fulfilled if the function in question is $d - 1$ times continuously differentiable. We now state our first global convergence result. It involves a condition on the initialization, a *separation* property, that can only be satisfied in the many-particle limit. In an ambient space $\Omega$, we say that a set $C$ *separates* the sets $A$ and $B$ if any continuous path in $\Omega$ with endpoints in $A$ and $B$ intersects $C$.

**Theorem 3.3.** *Under Assumptions 3.2, let $(\mu_t)_{t \geq 0}$ be a Wasserstein gradient flow of $F$ such that, for some $0 < r_a < r_b$, the support of $\mu_0$ is contained in $B(0, r_b)$ and separates the spheres $r_a \mathbb{S}^{d-1}$ and $r_b \mathbb{S}^{d-1}$. If $(\mu_t)_t$ converges to $\mu_\infty$ in $W_2$, then $\mu_\infty$ is a global minimizer of $F$ over $\mathcal{M}_+(\Omega)$. In particular, if $(\mathbf{u}_m(t))_{m \in \mathbb{N}, t \geq 0}$ is a sequence of classical gradient flows initialized in $B(0, r_b)$ such that $\mu_{m,0}$ converges weakly to $\mu_0$ then (limits can be interchanged)*

$$\lim_{t,m \to \infty} F(\mu_{m,t}) = \min_{\mu \in \mathcal{M}_+(\Omega)} F(\mu).$$

A proof and stronger statements are presented in Appendix C. There, we give a criterion for Wasserstein gradient flows to escape neighborhoods of non-optimal measures—also valid in the finite-particle

setting—and then show that it is always satisfied by the flow defined above. We also weaken the assumption that $\mu_t$ converges: we only need a certain projection of $\mu_t$ to converge weakly. Finally, the fact that limits in $m$ and $t$ can be interchanged is not anecdotal: it shows that the convergence is not conditioned on a relative speed of growth of both parameters.

This result might be easier to understand by drawing an informal distinction between (i) the structural assumptions which are instrumental and (ii) the technical conditions which have a limited practical interest. The initialization and the homogeneity assumptions are of the first kind. The Sard-type regularity is in contrast a purely technical condition: it is generally hard to check and known counter-examples involve artificial constructions such as the Cantor function [37]. Similarly, when there is compactness, a gradient flow that does not converge is an unexpected (in some sense adversarial) behavior, see a counter-example in [2]. We were however not able to exclude this possibility under interesting assumptions (see a discussion in Appendix C.5).

### 3.3 The partially $1$-homogeneous case

Similar results hold in the partially 1-homogeneous setting, which covers the lifted problems of Section 2.1 when $\phi$ is bounded (e.g., sparse deconvolution and neural networks with sigmoid activation).

**Assumptions 3.4.** *The domain is $\Omega = \mathbb{R} \times \Theta$ with $\Theta \subset \mathbb{R}^{d-1}$, $\Phi(w, \theta) = w \cdot \phi(\theta)$ and $V(w, \theta) = |w|\tilde{V}(\theta)$ where $\phi$ and $\tilde{V}$ are bounded, differentiable with Lipschitz differential. Moreover,*

*(i)* (smooth convex loss) *The loss $R$ is convex, differentiable with differential $dR$ Lipschitz on bounded sets and bounded on sublevel sets,*

*(ii)* (Sard-type regularity) *For all $f \in \mathcal{F}$, the set of regular values of $g_f : \theta \in \Theta \mapsto \langle f, \phi(\theta) \rangle + \tilde{V}(\theta)$ is dense in its range, and*

*(iii)* (boundary conditions) *The function $\phi$ behaves nicely at the boundary of the domain: either*

  *(a)* $\Theta = \mathbb{R}^{d-1}$ *and for all $f \in \mathcal{F}$, $\theta \in \mathbb{S}^{d-2} \mapsto g_f(r\theta)$ converges, uniformly in $C^1(\mathbb{S}^{d-2})$ as $r \to \infty$, to a function satisfying the Sard-type regularity, or*

  *(b)* $\Theta$ *is the closure of an bounded open convex set and for all $f \in \mathcal{F}$, $g_f$ satisfies Neumann boundary conditions (i.e., for all $\theta \in \partial\Theta$, $d(g_f)_\theta(\vec{n}_\theta) = 0$ where $\vec{n}_\theta \in \mathbb{R}^{d-1}$ is the normal to $\partial\Theta$ at $\theta$).*

With the family of nested sets $Q_r := [-r, r] \times \Theta$, $r > 0$, these assumptions imply Assumptions 2.1. The following theorem mirrors the statement of Theorem 3.3, but with a different condition on the initialization. The remarks after Theorem 3.3 also apply here.

**Theorem 3.5.** *Under Assumptions 3.4, let $(\mu_t)_{t\geq 0}$ be a Wasserstein gradient flow of $F$ such that for some $r_0 > 0$, the support of $\mu_0$ is contained in $[-r_0, r_0] \times \Theta$ and separates $\{-r_0\} \times \Theta$ from $\{r_0\} \times \Theta$. If $(\mu_t)_t$ converges to $\mu_\infty$ in $W_2$, then $\mu_\infty$ is a global minimizer of $F$ over $\mathcal{M}_+(\Omega)$. In particular, if $(\mathbf{u}_m(t))_{m\in\mathbb{N}, t\geq 0}$ is a sequence of classical gradient flows initialized in $[-r_0, r_0] \times \Theta$ such that $\mu_{m,0}$ converges to $\mu_0$ in $W_2$ then (limits can be interchanged)*

$$\lim_{t,m\to\infty} F(\mu_{m,t}) = \min_{\mu\in\mathcal{M}_+(\Omega)} F(\mu).$$

## 4 Case studies and numerical illustrations

In this section, we apply the previous abstract statements to specific examples and show on synthetic experiments that the particle-complexity to reach global optimality is very favorable.

### 4.1 Sparse deconvolution

For sparse deconvolution, it is typical to consider a signal $y \in \mathcal{F} := L^2(\Theta)$ on the $d$-torus $\Theta = \mathbb{R}^d/\mathbb{Z}^d$. The loss function is $R(f) = (1/2\lambda)\|y - f\|_{L^2}^2$ for some $\lambda > 0$, a parameter that increases with the noise level and the regularization is $V(w, \theta) = |w|$. Consider a filter impulse response $\psi : \Theta \to \mathbb{R}$ and let $\Phi(w, \theta) : x \mapsto w \cdot \psi(x - \theta)$. The object sought after is a signed measure on $\Theta$, which is obtained from a probability measure on $\mathbb{R} \times \Theta$ by applying a operator defined by $h_1(\mu)(B) = \int_\mathbb{R} w\mathrm{d}\mu(w, B)$ for all measurable $B \subset \Theta$. We show in Appendix D that Theorem 3.5 applies.

**Proposition 4.1** (Sparse deconvolution). *Assume that the filter impulse response $\psi$ is $\min\{2, d\}$ times continuously differentiable, and that the support of $\mu_0$ contains $\{0\} \times \Theta$. If the projection $(h^1(\mu_t))_t$ of the Wasserstein gradient flow of $F$ weakly converges to $\nu \in \mathcal{M}(\Theta)$, then $\nu$ is a global minimizer of*

$$\min_{\mu \in \mathcal{M}(\Theta)} \frac{1}{2\lambda} \left\| y - \int \psi \mathrm{d}\mu \right\|_{L^2}^2 + |\mu|(\Theta).$$

We show an example of such a reconstruction on the 1-torus on Figure 1, where the ground truth consists of $m_0 = 5$ weighted spikes, $\psi$ is an ideal low pass filter (a Dirichlet kernel of order 7) and $y$ is a noisy observation of the filtered spikes. The particle gradient flow is integrated with the forward-backward algorithm [11] and the particles initialized on a uniform grid on $\{0\} \times \Theta$.

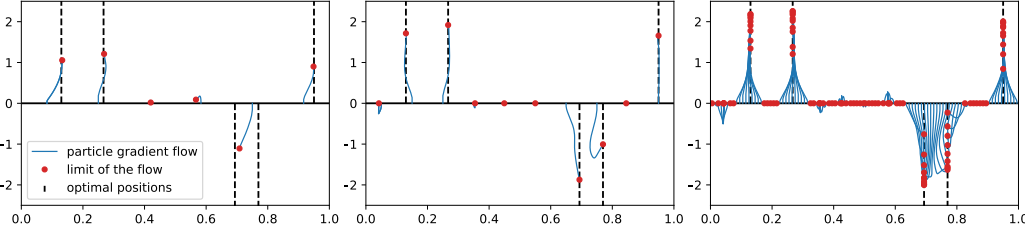

Figure 1: Particle gradient flow for sparse deconvolution on the 1-torus (horizontal axis shows positions, vertical axis shows weights). Failure to find a minimizer with 6 particles, success with 10 and 100 particles (an animated plot of this particle gradient flow can be found in Appendix D.5).

## 4.2 Neural networks with a single hidden layer

We consider a joint distribution of features and labels $\rho \in \mathcal{P}(\mathbb{R}^{d-2} \times \mathbb{R})$ and $\rho_x \in \mathcal{P}(\mathbb{R}^{d-2})$ the marginal distribution of features. The loss is the expected risk $R(f) = \int \ell(f(x), y) \mathrm{d}\rho(x, y)$ defined on $\mathcal{F} = L^2(\rho_x)$, where $\ell : \mathbb{R} \times \mathbb{R} \to \mathbb{R}_+$ is either the squared loss or the logistic loss. Also, we set $\Phi(w, \theta) : x \mapsto w\sigma(\sum_{i=1}^{d-2} \theta_i x_i + \theta_{d-1})$ for an activation function $\sigma : \mathbb{R} \to \mathbb{R}$. Depending on the choice of $\sigma$, we face two different situations.

**Sigmoid activation.** If $\sigma$ is a sigmoid, say $\sigma(s) = (1 + e^{-s})^{-1}$, then Theorem 3.5, with domain $\Theta = \mathbb{R}^{d-1}$ applies. The natural (optional) regularization term is $V(w, \theta) = |w|$, which amounts to penalizing the $\ell^1$ norm of the weights.

**Proposition 4.2** (Sigmoid activation). *Assume that $\rho_x$ has finite moments up to order $\min\{4, 2d-2\}$, that the support of $\mu_0$ is $\{0\} \times \Theta$ and that boundary condition 3.4-(iii)-(a) holds. If the Wasserstein gradient flow of $F$ converges in $W_2$ to $\mu_\infty$, then $\mu_\infty$ is a global minimizer of $F$.*

Note that we have to explicitly assume the boundary condition 3.4-(iii)-(a) because the Sard-type regularity at infinity cannot be checked *a priori* (this technical detail is discussed in Appendix D.3).

**ReLU activation.** The activation function $\sigma(s) = \max\{0, s\}$ is positively 1-homogeneous: this makes $\Phi$ 2-homogeneous and corresponds, at a formal level, to the setting of Theorem 3.3. An admissible choice of regularizer here would be the (semi-convex) function $V(w, \theta) = |w| \cdot |\theta|$ [4]. However, as shown in Appendix D.4, the differential $d\Phi$ has discontinuities: this prevents altogether from defining gradient flows, even in the finite-particle regime.

Still, a statement holds for a different parameterization of the *same class* of functions, which makes $\Phi$ differentiable. To see this, consider a domain $\Theta$ which is the disjoint union of 2 copies of $\mathbb{R}^d$. On the first copy, define $\Phi(\theta) : x \mapsto \sigma(\sum_{i=1}^{d-1} s(\theta_i)x_i + s(\theta_d))$ where $s(\theta_i) = \theta_i|\theta_i|$ is the signed square function. On the second copy, $\Phi$ has the same definition but with a minus sign. This trick allows to have the same expression power than classical ReLU networks. In practice, it corresponds to simply putting, say, random signs in front of the activation. The regularizer here can be $V(\theta) = |\theta|^2$.

**Proposition 4.3** (Relu activation). *Assume that $\rho_x \in \mathcal{P}(\mathbb{R}^{d-1})$ has finite second moments, that the support of $\mu_0$ is $r_0\mathbb{S}^{d-1}$ for some $r_0 > 0$ (on both copies of $\mathbb{R}^d$) and that the Sard-type regularity*

*Assumption 3.2-(ii) holds. If the Wasserstein gradient flow of $F$ converges in $W_2$ to $\mu_\infty$, then $\mu_\infty$ is a global minimizer of $F$.*

We display on Figure 2 particle gradient flows for training a neural network with a single hidden layer and ReLU activation in the classical (non-differentiable) parameterization, with $d = 2$ (no regularization). Features are normally distributed, and the ground truth labels are generated with a similar network with $m_0 = 4$ neurons. The particle gradient flow is "integrated" with mini-batch SGD and the particles are initialized on a small centered sphere.

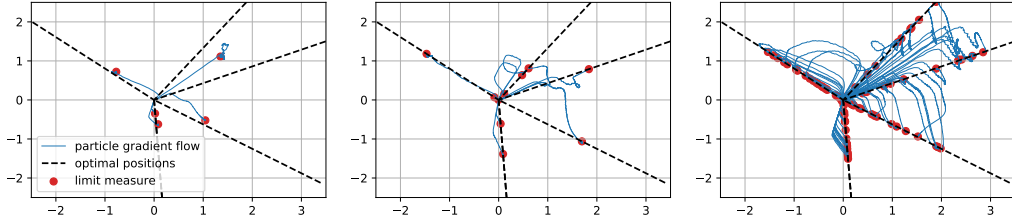

Figure 2: Training a neural network with ReLU activation. Failure with 5 particles (a.k.a. neurons), success with 10 and 100 particles. We show the trajectory of $|w(t)| \cdot \theta(t) \in \mathbb{R}^2$ for each particle (an animated plot of this particle gradient flow can be found in Appendix D.5).

### 4.3 Empirical particle-complexity

Since our convergence results are non-quantitative, one might argue that similar—and much simpler to prove—asymptotical results hold for the method of distributing particles on the whole of $\Theta$ and simply optimizing on the weights, which is a convex problem. Yet, the comparison of the particle-complexity shown in Figure 3 stands strongly in favor of particle gradient flows. While exponential particle-complexity is unavoidable for the convex approach, we observed on several synthetic problems that particle gradient descent only needs a slight over-parameterization $m > m_0$ to find global minimizers within optimization error (see details in Appendix D.5).

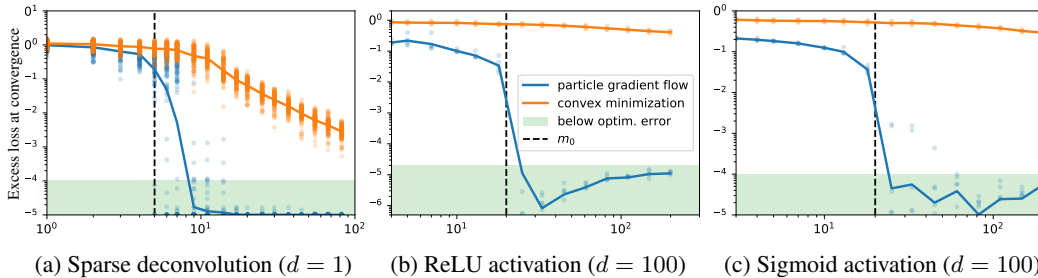

(a) Sparse deconvolution ($d = 1$)   (b) ReLU activation ($d = 100$)   (c) Sigmoid activation ($d = 100$)

Figure 3: Comparison of particle-complexity for particle gradient flow and convex minimization on a fixed grid: excess loss at convergence vs. number of particles. Simplest minimizer has $m_0$ particles.

## 5 Conclusion

We have established asymptotic global optimality properties for a family of non-convex gradient flows. These results were enabled by the study of a Wasserstein gradient flow: this object simplifies the handling of many-particle regimes, analogously to a mean-field limit. The particle-complexity to reach global optimality turns out very favorable on synthetic numerical problems. This confirms the relevance of our qualitative results and calls for quantitative ones that would further exploit the properties of such particle gradient flows. Multiple layer neural networks are also an interesting avenue for future research.

**Acknowledgments**

We acknowledge supports from grants from Région Ile-de-France and the European Research Council (grant SEQUOIA 724063).

## Footnotes

[1]A function $f : \mathbb{R}^d \to \mathbb{R}$ is semiconvex, or $\lambda$-convex, if $f + \lambda |\cdot|^2$ is convex, for some $\lambda \in \mathbb{R}$. On a compact domain, any smooth fonction is semiconvex.

[2]An absolutely continuous function $x : \mathbb{R} \to \mathbb{R}^d$ is almost everywhere differentiable and satisfies $x(t) - x(s) = \int_s^t x'(r)dr$ for all $s < t$.

[3]For a smooth vector field $E = (E_i)_{i=1}^d : \mathbb{R}^d \to \mathbb{R}^d$, its divergence is given by $\mathrm{div}(E) = \sum_{i=1}^d \partial E_i / \partial x_i$.

[4]For a function $g : \Theta \to \mathbb{R}$, a regular value is a real number $\alpha$ in the range of $g$ such that $g^{-1}(\alpha)$ is included in an open set where $g$ is differentiable and where $dg$ does not vanish.

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
