[Supplementary Material]

## Supplementary material

Supplementary material for the paper: "On the Global Convergence of Gradient Descent for Over-parameterized Models using Optimal Transport" authored by Lénaïc Chizat and Francis Bach (NIPS 2018).

This appendix is organized as follows:

- Appendix A: Introductory facts
- Appendix B: Many-particle limit and Wasserstein gradient flow
- Appendix C: Convergence to global minimizers
- Appendix D: Case studies and numerical experiments

# A   Introductory facts

## A.1   Tools from measure theory

In this paper, the term *measure* refers to a *finite* signed measure on $\mathbb{R}^d$, $d \geq 1$, endowed with its Borel $\sigma$-algebra. We write $\mathcal{M}(X)$ for the set of such measures concentrated on a measurable set $X \subset \mathbb{R}^d$. Hereafter, we gather some concepts and facts from measure theory that are used in the proofs.

**Variation of a signed measure.**   The Jordan decomposition theorem [10, Cor. 4.1.6] asserts that any finite signed measure $\mu \in \mathcal{M}(\mathbb{R}^d)$ can be decomposed as $\mu = \mu_+ - \mu_-$ where $\mu_+, \mu_- \in \mathcal{M}_+(\mathbb{R}^d)$. If $\mu_+$ and $\mu_-$ are chosen with minimal total mass, the *variation* of $\mu$ is the nonnegative measure $|\mu| := \mu_+ + \mu_-$ and $|\mu|(\mathbb{R}^d)$ is the *total variation norm* of $\mu$.

**Support and concentration set.**   The *support* spt $\mu$ of a measure $\mu \in \mathcal{M}_+(\mathbb{R}^d)$ is the complement of the largest open set of measure 0, or, equivalently, the set of points which neighborhoods have positive measure. We say that $\mu$ is *concentrated* on a set $S \subset \mathbb{R}^d$ if the complement of $S$ is included in a measurable set of measure 0. In particular, $\mu$ is concentrated on spt $\mu$.

**Pushforward.**   Let $X$ and $Y$ be measurable subsets of $\mathbb{R}^d$ and let $T : X \to Y$ be a measurable map. To any measure $\mu \in \mathcal{M}(X)$ corresponds a measure $T_{\#}\mu \in \mathcal{M}(Y)$ called the *pushfoward* of $\mu$ by $T$. It is defined as $T_{\#}\mu(B) = \mu(T^{-1}(B))$ for all measurable set $B \subset Y$ and corresponds to the distribution of the "mass" of $\mu$ after it has been displaced by the map $T$. It satisfies $\int_Y \varphi \, \mathrm{d}(T_{\#}\mu) = \int_X \varphi \circ T \, \mathrm{d}\mu$ whenever $\varphi : Y \to \mathbb{R}$ is a measurable function such that $\varphi \circ T$ is $\mu$-integrable [10, Prop. 2.6.8]. In particular, with a projection map $\pi^i : (x_1, x_2, \dots) \mapsto x_i$, the pushforward $\pi^i_{\#}\mu$ is the *marginal* of $\mu$ on the $i$-th factor.

**Weak convergence and Bounded Lipschitz norm.**   We say that a sequence of measures $\mu_n \in \mathcal{M}(\mathbb{R}^d)$ *weakly* (or *narrowly*) converges to $\mu$ if, for all continuous and bounded function $\varphi : \mathbb{R}^d \to \mathbb{R}$ it holds $\int \varphi \mathrm{d}\mu_n \to \int \varphi \mathrm{d}\mu$. For sequences which are bounded in total variation norm, this is equivalent to the convergence in Bounded Lipschitz norm. The latter is defined, for $\mu \in \mathcal{M}(\mathbb{R}^d)$, as

$$\|\mu\|_{\mathrm{BL}} := \sup \left\{ \int \varphi \, \mathrm{d}\mu \; ; \; \varphi : \mathbb{R}^d \to \mathbb{R}, \; \mathrm{Lip}(\varphi) \leq 1, \; \|\varphi\|_\infty \leq 1 \right\} \tag{7}$$

where $\mathrm{Lip}(\varphi)$ is the smallest Lipschitz constant of $\varphi$ and $\| \cdot \|_\infty$ the supremum norm.

**Wasserstein metric.**   The $p$-Wasserstein distance between two probability measures $\mu, \nu \in \mathcal{P}(\mathbb{R}^d)$ is defined as

$$W_p(\mu, \nu) := \left( \min \int |y - x|^p d\gamma(x, y) \right)^{1/p}$$

where the minimization is over the set of probability measures $\gamma \in \mathcal{P}(\mathbb{R}^d \times \mathbb{R}^d)$ such that the marginal on the first factor $\mathbb{R}^d$ is $\mu$ and is $\nu$ on the second factor. The set of probability measures with finite second moments endowed with the metric $W_2$ is a complete metric space that we denote $\mathcal{P}_2(\mathbb{R}^d)$. A sequence $(\mu_m)_m$ converges in $\mathcal{P}_2(\mathbb{R}^d)$ iff for all continuous function $\varphi : \mathbb{R}^d \to \mathbb{R}$ with at most

quadratic growth it holds $\int \varphi \mathrm{d}\mu_m \to \int \varphi \mathrm{d}\mu$ [3, Prop. 7.1.5] (this is stronger than weak convergence). Using, respectively, the duality formula for $W_1$ [29, Eq. (3.1)] and Jensen's inequality, it holds

$$\|\mu - \nu\|_{\mathrm{BL}} \le W_1(\mu, \nu) \le W_2(\mu, \nu).$$

Note that the functional of interest in this article is continuous for the Wasserstein metric. This strong regularity is rather rare in the study of Wasserstein gradient flows.

**Lemma A.1** (Wasserstein continuity of $F$). *Under Assumptions 2.1, the function $F$ is continuous for the Wasserstein metric $W_2$.*

*Proof.* Let $(\mu_m)_m, \mu \in \mathcal{P}_2(\Omega)$ be such that $W_2(\mu_m, \mu) \to 0$. By Assumption 2.1-(iii)-(c), $\|\Phi\|$ and $|V|$ have at most quadratic growth. It follows $\int V \mathrm{d}\mu_m \to \int V \mathrm{d}\mu$ and since, by the properties of Bochner integrals [10, Prop. E.5], it holds $\|\int \Phi \mathrm{d}\mu_m - \int \Phi \mathrm{d}\mu\| \le \int \|\Phi\| \mathrm{d}(\mu_m - \mu)$, we also have $\int \Phi \mathrm{d}\mu_m \to \int \Phi \mathrm{d}\mu$ strongly in $\mathcal{F}$. As $R$ is continuous in the strong topology of $\mathcal{F}$, it follows $F(\mu_m) \to F(\mu)$. $\qquad\square$

### A.2 Lifting to the space of probability measures

Let us give technical details about the lifting introduced in Section 2.1 that allows to pass from a problem on the space of signed measures on $\Theta \subset \mathbb{R}^{d-1}$ (the minimization of $J$ defined in (1)) to an equivalent problem on the space of probability measures on a bigger space $\Omega \subset \mathbb{R}^d$ (the minimization of $F$ defined in (3)).

**Homogeneity.** We recall that a function $f$ from $\mathbb{R}^d$ to a vector space is said *positively $p$-homogeneous*, with $p \ge 0$ if for all $u \in \mathbb{R}^d$ and $\lambda > 0$ it holds $f(\lambda u) = \lambda^p f(u)$. We often use without explicit mention the properties related to homogeneity such as the fact that the (sub)-derivative of a positively $p$-homogeneous function is positively $(p-1)$-homogeneous and, for $f$ differentiable (except possibly at 0), the identity $u \cdot \nabla f(u) = p f(u)$ for $u \ne 0$.

#### A.2.1 The partially 1-homogeneous case

We take $\Omega := \mathbb{R} \times \Theta$, $\Phi(w, \theta) = w \cdot \phi(\theta)$ and $V(w, \theta) = |w| \tilde{V}(\theta)$ for some continuous functions $\phi : \Theta \to \mathcal{F}$ and $\tilde{V} : \Theta \to \mathbb{R}_+$. This setting covers the lifted problems mentioned in Section 2.1. We first show that $F$ can be indifferently minimized over $\mathcal{M}_+(\Omega)$ or over $\mathcal{P}(\Omega)$, thanks to the homogeneity of $\Phi$ and $V$ in the variable $w$.

**Proposition A.2.** *For all $\mu \in \mathcal{M}_+(\Omega)$, there is $\nu \in \mathcal{P}(\Omega)$ such that $F(\mu) = F(\nu)$.*

*Proof.* If $|\mu|(\Omega) = 0$ then $F(\mu) = 0 = F(\delta_{(0,\theta_0)})$ where $\theta_0$ is any point in $\Theta$. Otherwise, we define the map $T : (w, \theta) \mapsto (|\mu|(\Omega) \cdot w, \theta)$ and the probability measure $\nu := T_\#(\mu/|\mu|(\Omega)) \in \mathcal{P}(\Omega)$, which satisfies $F(\nu) = F(\mu)$. $\qquad\square$

We now introduce a projection operator $h^1 : \mathcal{M}_+(\Omega) \to \mathcal{M}(\Theta)$ that is adapted to the partial homogeneity of $\Phi$ and $V$. It is defined by $h^1(\mu)(B) = \int_\mathbb{R} w \mu(dw, B)$ for all $\mu \in \mathcal{P}(\Omega)$ and measurable set $B \subset \Theta$ or, equivalently, by the property that for all continuous and bounded test function $\varphi : \Theta \to \mathbb{R}$,

$$\int_\Theta \varphi(\theta) \mathrm{d}h^1(\mu)(\theta) = \int_{\mathbb{R} \times \Theta} w \varphi(\theta) d\mu(w, \theta).$$

This operator is well defined whenever $(w, \theta) \mapsto w$ is $\mu$-integrable.

**Proposition A.3** (Equivalence under lifting). *It holds $\mathcal{M}(\Theta) \subset h^1(\mathcal{P}(\Omega)) = h^1(\mathcal{M}_+(\Omega))$. For a regularizer $G$ on $\mathcal{M}(\Theta)$ of the form $G(\mu) = \inf_{\nu \in h^{-1}(\mu)} \int_\Omega V d\nu$, it holds $\inf_{\nu \in \mathcal{M}(\Theta)} J(\nu) = \inf_{\mu \in \mathcal{M}_+(\Omega)} F(\mu)$. If the infimum defining $G$ is attained and if $\nu \in \mathcal{M}(\Theta)$ minimizes $J$, then there exists $\mu \in h^{-1}(\nu)$ that minimizes $F$ over $\mathcal{M}_+(\Omega)$.*

*Proof.* A signed measure $\nu \in \mathcal{M}(\Theta)$ can be expressed as $\nu = f\sigma$ where $\sigma \in \mathcal{P}(\Theta)$ and $f : \Theta \to \mathbb{R} \in L^1(\sigma)$ (take for instance $\sigma$ the normalized variation of $\mu$ if $|\mu|(\Theta) > 0$). The measure

$$\mu := (f \times \mathrm{id})_\# \sigma \qquad (8)$$

belongs to $\mathcal{P}(\Omega)$ and satisfies $h^1(\nu) = \mu$. This proves that $h^1(\mathcal{P}(\Omega))$ is surjective. It is clear by the definition of $h^1$ that for all $\mu \in h^{-1}(\nu)$, it holds $\int \Phi \, d\mu = \int \phi \, d\nu$ hence $F(\mu) \geq J(\nu)$, with equality when $\mu$ is the minimizer in the definition of $G$. $\qquad\square$

The class of regularizer considered in Proposition A.3 includes the total variation norm.

**Proposition A.4** (Total variation). *Let $V(w, \theta) = |w|$. For $\mu \in \mathcal{M}(\Theta)$, it holds $\int V \, d\mu \geq |h^1(\mu)|(\Theta)$ with equality if, for instance, $\mu$ is a lift of $h^1(\mu)$ of the form* (8).

*Proof.* Let $\mu \in \mathcal{P}(\Omega)$ and $\nu = h^1(\mu)$. We define $\tilde{\nu}_+ := \int_{\mathbb{R}_+} w \, \mu(w, \cdot)$ and $\tilde{\nu}_- := -\int_{\mathbb{R}_-} w \, \mu(w, \cdot)$. Clearly, $\nu = \tilde{\nu}_+ - \tilde{\nu}_-$ and by the definition of the total variation of a signed measure, $|\nu|(\Theta) = |\nu_+|(\Theta) + |\nu_-|(\Theta) \leq |\tilde{\nu}_+|(\Theta) + |\tilde{\nu}_-|(\Theta) = \int V \, d\mu$. There is equality whenever $\operatorname{spt} \tilde{\nu}_+ \cap \operatorname{spt} \tilde{\nu}_-$ has $|\nu|$-measure 0 (see [10, Cor. 4.1.6]), a condition which is satisfied by the lift in (8). $\qquad\square$

### A.2.2   The $2$-homogeneous case

Another structure that is studied in this paper is when $\Phi$ and $V$ are defined on $\mathbb{R}^d$ and are positively 2-homogeneous. In this case, the role played by $\Theta$ is the previous section is played by the unit sphere $\mathbb{S}^{d-1}$ of $\mathbb{R}^d$. We could again make links between $F$ (defined as in Eq. (3)) and a functional on nonnegative measures on the sphere (playing the role of $J$) but here we will limit ourselves to defining the projection operator relevant in this setting. It is $h^2 : \mathcal{M}_+(\mathbb{R}^d) \to \mathcal{M}_+(\mathbb{S}^{d-1})$ characterized by the relationship, for all continuous and bounded function $\varphi : \mathbb{S}^{d-1} \to \mathbb{R}$ (with the convention $\phi(0/0) = 0$):

$$\int_{\mathbb{S}^{d-1}} \varphi(\theta) \mathrm{d}h^2(\mu)(\theta) = \int_{\mathbb{R}^d} |u|^2 \varphi(u/|u|) d\mu(u).$$

This operator is well-defined iff $\mu$ has finite second order moments.

## B   Many-particle limit and Wasserstein gradient flow

### B.1   Proof of Proposition 2.3

As the sum of a continuously differentiable and a semiconvex function, $F_m$ is locally semiconvex and the existence of a unique gradient flow on a maximal interval $[0, T[$ with the claimed properties is standard, see [30, Sec. 2.1]. Now, a general property of gradient flows is that for a.e $t \in \mathbb{R}_+, u \in \Omega$, the derivative is (minus) the subgradient of minimal norm. This leads to the explicit formula involving the velocity field with pointwise minimal norm:

$$\begin{aligned}
v_t(u) &= \arg\min \left\{ |v|^2 \; ; \; \tilde{v}_t(u) - v \in \partial V(u) \right\} \\
&= \tilde{v}_t(u) - \arg\min \left\{ |\tilde{v}_t(u) - z|^2 \; ; \; z \in \partial V(u) \right\} \\
&= (\mathrm{id} - \mathrm{proj}_{\partial V(u)})(\tilde{v}_t(u)).
\end{aligned}$$

In the specific case of gradient flows of lower bounded functions, we can derive estimates that imply that $T = \infty$ (even if $F_m$ is not globally semiconvex). Indeed, for all $t > 0$, it holds

$$F_m(\mathbf{u}(0)) - F_m(\mathbf{u}(t)) = -\int_0^t \frac{d}{ds} F_m(\mathbf{u}(s)) \mathrm{d}s = \frac{1}{m} \int_0^t |\mathbf{u}'(s)|^2 \mathrm{d}s \geq \frac{t}{m} \left( \int_0^t |\mathbf{u}'(s)| \mathrm{d}s \right)^2$$

by Jensen's inequality. Since $F_m$ is lower bounded, this proves that the gradient flow has bounded length on bounded time intervals. By compactness, if $T$ was finite then $\mathbf{u}(T)$ would exist, thus contradicting the maximality of $T$, hence $T = \infty$ and the gradient flow is globally defined.

### B.2   Link between classical and Wasserstein gradient flows

We first give a rigorous definition of the continuity equation which appear in the definition of Wasserstein gradient flows (Definition 2.4).

**Continuity equation.** Considerations from fluid mechanics suggest that if a time dependent distribution of mass $(\mu_t)_t$ is displaced under the action of a velocity field $(v_t)_t$, then the *continuity equation* is satisfied: $\partial_t \mu_t = -\text{div}(v_t \mu_t)$. For distributions which do not have a smooth density, this equation should be understood distributionally, which means that for all smooth test functions $\varphi : ]0, \infty[ \times \mathbb{R}^d$ with compact support, it holds

$$\int_0^\infty \int_{\mathbb{R}^d} (\partial_t \varphi_t(u) + \nabla_u \varphi_t(u) \cdot v_t(u)) \, d\mu_t(u) dt = 0.$$

The integrability condition $\int_0^{t_0} \int_{\mathbb{R}^d} |v_t(u)| d\mu(u) dt < \infty$ for all $t_0 < T$ should also hold.

As we show now, there is a precise link between classical and Wasserstein gradient flow (Definitions 2.2 and 2.4). This is a simple result but might be instructive for readers who are not familiar with the concept of distributional solutions of partial differential equations.

**Proposition B.1** (Atomic solutions of Wasserstein gradient flow). *If $\mathbf{u} : \mathbb{R}_+ \to \Omega^m$ is a classical gradient flow for $F_m$ in the sense of Definition 2.2, then $t \mapsto \mu_{m,t} := \frac{1}{m} \sum_{i=1}^m \delta_{\mathbf{u}_i(t)}$ is a Wasserstein gradient flow of $F$ in the sense of Definition 2.4.*

*Proof.* Let us call $v_t$ the velocity vector field defined in (5). In it easy to see that $t \mapsto \mu_{m,t}$ is absolutely continuous for $W_2$ and for any smooth function with compact support $\varphi : ]0, \infty[ \times \mathbb{R}^d \to \mathbb{R}$, we have

$$0 = \frac{1}{m} \sum_{i=1}^m \int_{\mathbb{R}_+} \frac{d}{dt} \varphi_t(\mathbf{u}_i(t)) \, dt$$

$$= \frac{1}{m} \sum_{i=1}^m \int_{\mathbb{R}_+} (\partial_t \varphi_t(\mathbf{u}_i(t)) + \nabla_u \varphi_t(\mathbf{u}_i(t)) \cdot v_t(\mathbf{u}_i)) \, dt$$

$$= \int_{\mathbb{R}_+} \int_\Omega (\partial_t \varphi_t(u) + \nabla_u \varphi_t(u) \cdot v_t(u)) \, d\mu_{m,t} dt$$

which precisely means that $(\mu_{m,t})_{t \geq 0}$ is a distributional solution to (6). $\square$

Note that $(\mu_{m,t})_t$ has the same number of atoms throughout the dynamic. In particular, if no minimizer of $F$ is an atomic measure with at most $m$ atoms, then $(\mu_{m,t})_t$ is guaranteed to *not* converge to a minimizer.

### B.2.1 Properties of the Wasserstein gradient flow (proof of Proposition 2.5)

In this section, we use the general theory of Wasserstein gradient flows developed in [3] to prove existence and uniqueness of Wasserstein gradient flows as claimed in Proposition 2.5, under Assumptions 2.1. The "existence" part of the proof is in fact redundant with Theorem 2.6 which provides with another constructive proof. We recall that $F'(\mu) : \Omega \to \mathbb{R}$ is defined as

$$F'(\mu)(u) = \langle R'(\int \Phi d\mu), \Phi(u) \rangle + V(u)$$

and that the field of subgradients of minimal norm of $F'(\mu)$ has an explicit formula given in (5). Our strategy is to use, as an intermediary step, the Wasserstein gradient flows for the family of functionals $F^{(r)} : \mathcal{P}_2(\Omega) \to \mathbb{R}$ defined, for $r > 0$, as

$$F^{(r)}(\mu) = \begin{cases} F(\mu) & \text{if } \mu(Q_r) = 1, \\ \infty & \text{otherwise} \end{cases}$$

where $(Q_r)_{r>0}$ is the nested family of subsets of $\mathbb{R}^d$ that appear in Assumptions 2.1. These "localized" functionals have nice properties in the Wasserstein geometry, as shown in Lemma B.2. For $r > 0$, we say that $\gamma \in \mathcal{P}(\Omega \times \Omega)$ is an *admissible transport plan* if both its marginals are concentrated on $Q_r$ and have finite second moments. The transport cost associated to $\gamma$ is denoted $C_p(\gamma) := \left( \int |y - x|^p d\gamma(x, y) \right)^{1/p}$ for $p \geq 1$, and we introduce the quantities

$$\|d\Phi\|_{\infty,r} = \sup_{u \in Q_r} \|d\Phi_u\| \qquad\qquad L_{d\Phi} = \sup_{\substack{u, \tilde{u} \in Q_r \\ u \neq \tilde{u}}} \frac{\|d\Phi_{\tilde{u}} - d\Phi_u\|}{|\tilde{u} - u|}$$

$$\|dR\|_{\infty,r} = \sup_{f \in \mathcal{F}_r} \|dR_f\| \qquad\qquad L_{dR} = \sup_{\substack{f, g \in \mathcal{F}_r \\ f \neq g}} \frac{\|dR_f - dR_g\|}{\|f - g\|}$$

where $\mathcal{F}_r := \{\int \Phi \mathrm{d}\mu \; ; \; \mu \in \mathcal{P}(\Omega), \, \mu(Q_r) = 1\}$ is bounded in $\mathcal{F}$. Those quantities are finite for all $r > 0$ under Assumptions 2.1. For the sake of clarity, we set $V = 0$ in the next lemma, and focus on the *loss* term, which is a new object of study. The term involving $V$, well-studied in the theory of Wasserstein gradient flows (see [3, Prop. 10.4.2]), is incorporated later.

**Lemma B.2** (Properties of $F^{(r)}$ in Wasserstein geometry). *Under Assumptions 2.1, suppose that $V = 0$. For all $r > 0$, $F^{(r)}$ is proper and continuous for $W_2$ on its closed domain. Moreover,*

  (i) *there exists $\lambda_r > 0$ such that for all admissible transport plan $\gamma$, considering the transport interpolation $\mu_t^\gamma := ((1-t)\pi^1 + t\pi^2)_{\#}\gamma$, the function $t \mapsto F(\mu_t^\gamma)$ is differentiable with a $\lambda_r C_2^2(\gamma)$-Lipschitz derivative;*

 (ii) *for $\mu$ concentrated on $Q_r$, a velocity field $v \in L^2(\mu, \mathbb{R}^d)$ satisfies, for any admissible transport plan $\gamma$ with first marginal $\mu$,*

$$F(\pi_{\#}^2 \gamma) \geq F(\mu) + \int v(u) \cdot (\tilde{u} - u) \mathrm{d}\gamma(u, \tilde{u}) + o(C_2(\gamma))$$

*if and only if $v(u) \in \partial(F'(\mu) + \iota_{Q_r})(u)$ for $\mu$-almost every $u \in \Omega$, where $\iota_{Q_r}$ is the convex function on $\Omega$ that is worth $0$ on $Q_r$ and $\infty$ outside.*

*Proof.* First, it is clear that $F$ is proper because $F^{(r)}(\delta_{u_0}) = R(\Phi(u_0))$ is finite whenever $u_0 \in Q_r$. It is moreover continuous (see Lemma A.1) on its closed domain $\{\mu \in \mathcal{P}_2(\Omega) \; ; \; \mu(Q_r) = 1\}$.

**Proof of (i).** Let us denote $h(t) := F^{(r)}(\mu_t^\gamma)$. Since $dR$ and $d\Phi$ are Lipschitz on $\mathcal{F}_r$ and $Q_r$ respectively, $h(t)$ is differentiable with

$$h'(t) = \frac{d}{dt} F^{(r)}(\mu_t^\gamma) = \left\langle R'(\int \Phi \mathrm{d}\mu_t^\gamma), \int d\Phi_{(1-t)x+ty}(y-x)\mathrm{d}\gamma(x,y) \right\rangle.$$

In particular, we can differentiate $t \mapsto \int \Phi \mathrm{d}\mu_t^\gamma = \int \Phi((1-t)x + ty)\mathrm{d}\gamma(x,y)$ because all $(\mu_t^\gamma)_t$ are supported on $Q_r$ where $d\Phi$ is uniformly bounded and Bochner integrals admit a dominated convergence theorem [10, Thm. E6]. For $0 \leq t_1 < t_2 < 1$, we have the bounds

$$|h'(t_2) - h'(t_1)| \leq (I) + (II)$$

where, one the one hand,

$$
\begin{aligned}
(I) &= \left| \left\langle R'\left(\int \Phi \mathrm{d}\mu_{t_2}\right) - R'\left(\int \Phi \mathrm{d}\mu_{t_1}\right), \int d\Phi_{(1-t_2)x+t_2 y}(y-x)\mathrm{d}\gamma(x,y) \right\rangle \right| \\
&\leq [L_{dR} \cdot \|d\Phi\|_{\infty,r} \cdot |t_2 - t_1| \cdot C_1(\gamma)] \cdot [\|d\Phi\|_{\infty,r} \cdot C_1(\gamma)] \\
&\leq L_{dR} \cdot \|d\Phi\|_{\infty,r}^2 \cdot C_2^2(\gamma) \cdot |t_2 - t_1|
\end{aligned}
$$

where we used Hölder's inequality to obtain $C_1^2(\gamma) \leq C_2^2(\gamma)$ is the last line. On the other hand,

$$
\begin{aligned}
(II) &= \left| \left\langle R'\left(\int \Phi \mathrm{d}\mu_{t_1}\right), \int \left[d\Phi_{(1-t_2)x+t_2 y} - d\Phi_{(1-t_1)x+t_1 y}\right](y-x)\mathrm{d}\gamma(x,y) \right\rangle \right| \\
&\leq L_{d\Phi} \cdot \|dR\|_{\infty,r} \cdot C_2^2(\gamma) \cdot |t_2 - t_1|.
\end{aligned}
$$

As a consequence, $h'$ is $\lambda_r \cdot C_2^2(\gamma)$ Lipschitz with $\lambda_r = L_{dR}\|d\Phi\|_{\infty,r}^2 + L_{d\Phi}\|dR\|_{\infty,r}$. In particular, using the notions defined in [3], $F^{(r)}$ is $(-\lambda_r)$-geodesically semiconvex. Remark that these bounds may explode when $r$ goes to infinity: this explains why we work with measures supported on $Q_r$.

**Proof of (ii).** The proof is similar, with the difference that this property is a local one. We have the first-order Taylor expansions, for $u, \tilde{u} \in Q_r$ and $f, g \in \mathcal{F}_r$,

$$
\begin{aligned}
\Phi(\tilde{u}) &= \Phi(u) + d\Phi(\tilde{u} - u) + M(u, \tilde{u}) \\
R(g) &= R(f) + \langle R'(f), g - f \rangle + N(f, g)
\end{aligned}
$$

where the remainders $M$ and $N$ satisfy $\|M(u, \tilde{u})\| \leq \frac{1}{2} L_{d\Phi} \cdot |\tilde{u} - u|^2$ and $\|N(f,g)\| \leq \frac{1}{2} L_{dR} \cdot \|g - f\|^2$. We denote by $\mu$ and $\nu$ the first and second marginals of $\gamma$, assume that they are both concentrated on $Q_r$, and obtain, by composition, the Taylor expansion

$$F^{(r)}(\nu) = F^{(r)}(\mu) + \left\langle R'(\int \Phi \mathrm{d}\mu), \int d\Phi_u(\tilde{u} - u)\mathrm{d}\gamma(u, \tilde{u}) \right\rangle + (I) + (II)$$

where $(I) = \langle R'(\int \Phi d\mu), \int M(u, \tilde{u}) d\gamma(u, \tilde{u}) \rangle$, so

$$|(I)| \leq \frac{1}{2} \|dR\|_{\infty, r} \cdot L_{d\Phi} \cdot C_2^2(\gamma) = o(C_2(\gamma))$$

and $(II) = N(\int \Phi d\mu, \int \Phi d\nu)$, so

$$|(II)| \leq \frac{1}{2} L_{dR} \cdot \| \int d\Phi_u(\tilde{u} - u) d\gamma(u, \tilde{u}) + \int M(u, \tilde{u}) d\gamma(u, \tilde{u})\|^2$$

$$\leq \frac{1}{2} L_{dR} \cdot \left( \|d\Phi\|_{\infty, r} \cdot C_1(\gamma) + \frac{1}{2} \cdot L_{d\Phi} \cdot C_2^2(\gamma) \right)^2 = o(C_2(\gamma))$$

where we used Hölder's inequality for the bound $C_1^2(\gamma) \leq C_2^2(\gamma)$. As a consequence,

$$F^{(r)}(\nu) = F^{(r)}(\mu) + \int \langle R'(\int \Phi d\mu), d\Phi_u(\tilde{u} - u) \rangle d\gamma(u, \tilde{u}) + o(C_2(\gamma))$$

and remember that the $j$-th component of $\nabla F'(\mu)$ is $u \mapsto \langle R'(\int \Phi d\mu), d\Phi_u(e_j) \rangle$ where $e_j$ is the $j$-th vector of the canonical basis of $\mathbb{R}^d$. This completely characterizes a velocity field satisfying (ii) on the interior of $Q_r$. On the boundary of $Q_r$, there is more freedom in the choice of $v(u)$ since $\pi_\#^2 \gamma$ is constrained to be supported on $Q_r$ so $v(u) - \nabla F'(\mu)(u)$ can live in the normal cone of $Q_r$ at $u$, which is the set $\partial \iota_{Q_r}(u)$. The condition thus relaxes as $v(u) \in \partial(F'(\mu) + \iota_{Q_r})(u)$. □

The previous properties are sufficient to guarantee that Wasserstein gradient flows for the functionals $F^{(r)}$ are well defined.

**Lemma B.3.** *Under Assumptions 2.1, there exists a unique Wasserstein gradient flow for $F^{(r)}$ starting from any $\mu_0 \in \mathcal{P}_2(\Omega)$ concentrated on $Q_r$, i.e. a curve $(\mu_t^{(r)})_{t \geq 0}$, continuous in $\mathcal{P}_2(\Omega)$, that solves $\partial_t \mu_t^{(r)} + \mathrm{div}(v_t^{(r)} \mu_t^{(r)}) = 0$ where, for all $t > 0$, $v_t^{(r)}(u) \in \partial(F'(\mu_t^{(r)})(u) + \iota_{Q_r}(u))$ for $\mu_t^{(r)}$-a.e $u \in \Omega$.*

*Proof.* It is easy to see that if $V$ is $\lambda_V$-semiconvex, then the function $\mu \mapsto \int V d\mu$ is $\lambda_V$ semiconvex along generalized geodesics (in the sense of [3, Def. 9.2.4], see [3, Prop. 10.4.2]). Combining with Lemma B.2-(i), we have that $F^{(r)}$ is $(\lambda_V - \lambda_r)$-semiconvex along generalized geodesics. Moreover, Lemma B.2-(ii) implies that $F^{(r)}$ admits strong Wasserstein subdifferentials on its domain [3, Def 10.3.1] and again, it is an easy adaptation to show that (ii) still holds with a potential term. So the existence of a unique Wasserstein gradient flow characterized as above is guaranteed by [3, Thm. 11.2.1]. □

We are in position to prove the well-posedness of Wasserstein gradient flows for the original functional $F$. Notice that, by the characterization in Lemma B.3, the Wasserstein gradient flows for the functions $F^{(r)}$ all coincide for $r > r_0 > 0$ on $[0, T]$ if $\mu_t^{(2r_0)}$ is concentrated in $Q_{r_0}$ for all $t \in [0, T]$. Our strategy is thus simply to make sure that for all time $T$, such a $r_0 > 0$ exists, i.e. to make sure that the support of gradient flows does not grow too fast.

*Proof of Proposition 2.5.* Let $r_0$ be such that $\mu_0$ is concentrated on $Q_{r_0}$. Given Lemma B.3, for all $r > r_0$, there exists a unique, globally defined, Wasserstein gradient flow $(\mu_t^{(r)})_{t \geq 0}$ for $F^{(r)}$. For all $r > r_0$, consider the first exit time from $Q_r$:

$$t_r := \inf\{t > 0 ; \mu_t^{(2r)}(Q_r) < 1\}.$$

Note that the definition of $t_r$ involves the flow $(\mu_t^{2r})_t$ but in fact, for all $\bar{r} > r$ and $0 \leq t \leq t_r$, it holds $\mu_t^{(2r)} = \mu_t^{(\bar{r})}$ by the uniqueness in Lemma B.3. Thus, if $t_r > 0$, we have existence and uniqueness of a Wasserstein gradient flow in the sense of Definition 2.4 on $[0, t_r]$. It only remains to show that $\lim_{r \to \infty} t_r = \infty$ so that the gradient flow can be defined at all times.

Given the property of $v^{(r)}$ in Lemma B.2-(ii), for all time $0 \leq t \leq t_r$, it holds $v_t^{(r)} \in \partial F'(\mu_t^{(r)})$ in $L^2(\mu_t^{(r)}; \mathbb{R}^d)$. Therefore, using Assumption 2.1-(iii)-(c) and the boundedness of $dR$ on sublevel sets, we have the bound, for $0 \leq t \leq t_r$,

$$|v_t^{(r)}(u)| \leq C_1 + C_2 r$$

with constants $C_1$ and $C_2$ independent of $u, r$ and $t$. This shows, by Grönwall's lemma applied to the flow of characteristics of the velocity field (this flow is defined below in Lemma B.4), that $\mu_t^{(r)}$ is concentrated on $\{u \in \Theta \; ; \; \text{dist}(u, Q_{r_0}) \leq (r_0 + C_1/C_2)e^{tC_2}\}$ and thus, for all $T > 0$ there exists $r > 0$ such that $t_r > T$. Hence $\lim_{r \to \infty} t_r = \infty$ and the gradient flow from Definition 2.4 is uniquely well-defined on $[0, T[$ for $T > 0$ arbitrary large. $\qquad \square$

Let us now add a useful representation lemma for the Wasserstein gradient flow as the pushforward of $\mu_0$ by the flow of the velocity fields.

**Lemma B.4** (Representation of the flow). *Under the assumptions of Proposition 2.5, let $(\mu_t)_{t \geq 0}$ be the Wasserstein gradient flow of $F$ and $(v_t)_t$ the associated velocity fields. Consider the flow $X : \mathbb{R}_+ \times \Omega \to \Omega$ which for all $u \in \Omega$, is an absolutely continuous solution to*

$$X(0, u) = u \quad and \quad \partial_t X(t, u) = v_t(X(t, u)) \text{ for a.e. } t \geq 0.$$

*Then $X$ is uniquely well-defined, continuous, $X(t, \cdot)$ is Lipschitz on $Q_r$, uniformly on compact time intervals for all $r > 0$, and it holds $\mu_t = (X_t)_{\#}\mu_0$.*

*Proof.* The claims concerning $X$ are classical and follow from the fact that $v_t$ satisfies a one-sided Lispchitz property on $Q_r$, uniformly on compact time intervals [3, Lemma 8.1.4]. The expression as a pushforward is also a general property of the continuity equation, see [3, Prop. 8.1.8]. $\qquad \square$

## B.3 Proof of the many-particle limit (Theorem 2.6)

While we could rely on abstract stability results for Wasserstein gradient flows [3, Thm.11.2.1 (Stability)] our proof is direct and uses basic arguments. It also gives an independent argument for the existence of Wasserstein gradient flows, distinct from the standard one : it involves a discretization in space instead of the classical discretization in time.

**Step (i).** We first show that, at least on a small time interval $[0, t_r]$, the paths are contained in $Q_r$ for some $r > r_0$. Let us introduce $t_r$ the first exit time from $Q_r$

$$t_r := \inf \{t > 0 \; ; \; \exists m \in \mathbb{N}, \mu_{m,t}(Q_r) < 1\}.$$

In order to show that $t_r$ is strictly positive, it is sufficient to bound the velocity of individual particles before $t_r$. Consider $L_{V,r}$ the Lipschitz constant of $V$ on $Q_r$. Given the expression of the velocity of each particle (given in Eq. (5)) and the minimum travel distance $r - r_0$ required to exit $Q_r$, we obtain the lower bound on the exit time $t_r \geq (r - r_0)/(\|d\Phi\|_{\infty,r}\|dR\|_{\infty,r} + L_{V,r}) > 0$.

**Step (ii).** Let us now work on the time interval $[0, t_r]$ and prove the existence of a limit curve $t \mapsto \mu_t$ in the space $\mathcal{P}_2(\Theta)$ using standard estimates for gradient flows and compactness. Our starting point is the bound, for $0 \leq t_1 < t_2 \leq t_r$,

$$W_2(\mu_{m,t_1}, \mu_{m,t_2})^2 \leq \frac{1}{m} \sum_{i=1}^{m} |\mathbf{u}_{m,i}(t_2) - \mathbf{u}_{m,i}(t_1)|^2 \leq \frac{(t_2 - t_1)}{m} \int_{t_1}^{t_2} \sum_{i=1}^{m} |\mathbf{u}'_{m,i}(s)|^2 \mathrm{d}s$$

which follows by matching each particle at $t_1$ to its future position at $t_2$, and by Jensen's inequality. Recalling the identity $\frac{1}{m} \sum_{i=1}^{m} |\mathbf{u}'_{m,i}(t)|^2 = -\frac{d}{dt} F(\mu_{m,t})$ from Proposition 2.3, it follows

$$W_2(\mu_{m,t_1}, \mu_{m,t_2})) \leq \sqrt{t_2 - t_1} \left( \sup_m F(\mu_{m,0}) - \inf_{\mu \in \mathcal{P}(\mathbb{R}^d)} F(\mu) \right)^{1/2}$$

and thus the family of curves $(t \mapsto \mu_{m,t})_m$ is equicontinuous in $W_2$ on $[0, t_r]$, uniformly in $m$. Moreover, for all $t \in [0, t_r]$, the family $(\mu_{m,t})_m$ lies in a $W_2$ ball, as such weakly precompact (but a priori not $W_2$-precompact). Since the weak topology is weaker than the topology of $W_2$, by Ascoli theorem, we can extract a subsequence converging weakly to a curve $(\mu_t)_{t \geq 0}$ continuous in the weak topology, which is concentrated in $Q_r$ at all time. We have also uniform convergence in the Bounded Lipschitz metric, which metrizes weak convergence of probability measures. In the following we only consider this subsequence, still denoted by $(\mu_m)_m$.

**Step (iii).**   The next step is to show that the limit curve $(\mu_t)$ satisfies a continuity equation as in Definition 2.4. Consider the velocity fields $v_{m,t}$ defined in Equation (5) and let us define $v_t$ the analog for the limit curve $(\mu_t)_t$. We want to show that the sequence $(E_m)_m$ of *momenta*, the vector valued measures on $[0, t_r] \times \Omega$ defined by $E_m := v_{m,t}\mu_{m,t}\mathrm{d}t$, converges weakly to $E := v_t\mu_t\mathrm{d}t$. Notice that these measures are also concentrated on $Q_r$. For any bounded and continuous function $\varphi : [0, t_r] \times \mathbb{R}^d \to \mathbb{R}^d$, it holds

$$\left| \int \varphi \cdot \mathrm{d}(E_m - E) \right| \leq \|\varphi\|_\infty \int |v_{m,t}(u) - v_t(u)| \mathrm{d}\mu_{m,t}(u)\mathrm{d}t + \left| \int \varphi \cdot v_t \mathrm{d}(\mu_{m,t} - \mu_t)\mathrm{d}t \right|. \quad (9)$$

We first prove that the first term in (9) tends to 0. Since all $(\mu_{m,t})_{m,t}$ are concentrated on $Q_r$, it is sufficient to show that the sequence of velocity fields $(t, u) \mapsto v_{m,t}(u)$ converges uniformly on $[0, t_r] \times Q_r$ to $(t, u) \mapsto v_t(u)$. We have, using the fact that a projection on a convex set is 1-Lipschitz,

$$|v_{m,t}(u) - v_t(u)| \leq 2|\tilde{v}_{t,m}(u) - \tilde{v}_t(u)| \leq 2\|d\Phi\|_{\infty,r} \cdot \|R'(\int \Phi \mathrm{d}\mu_{m,t}) - R'(\int \Phi \mathrm{d}\mu_t)\|.$$

Moreover, we have for all $t \in [0, t_r]$,

$$\begin{aligned}
\|R'(\int \Phi \mathrm{d}\mu_{m,t}) - R'(\int \Phi \mathrm{d}\mu_t)\| &\leq \|dR\|_{\infty,r} \cdot \|\int \Phi \mathrm{d}\mu_{m,t} - \int \Phi \mathrm{d}\mu_t\| \\
&\leq \|dR\|_{\infty,r} \cdot \sup_{f \in \mathcal{F}, \|f\| \leq 1} \int \langle f, \Phi(u) \rangle \mathrm{d}(\mu_{m,t} - \mu_t)(u) \\
&\leq \|dR\|_{\infty,r} \cdot \max\{\|\Phi\|_{\infty,r}, \|d\Phi\|_{\infty,r}\} \cdot \|\mu_{m,t} - \mu_t\|_{\mathrm{BL}}.
\end{aligned}$$

Since the convergence of $(t \to \mu_{m,t})_m$ is uniform in the Bounded Lipschitz norm, this proves uniform convergence of the velocity fields and the convergence of the first term in (9) to 0. The second term also converges to 0 because $(t, u) \mapsto \varphi(t, u) \cdot v_t(u)$ is continuous and bounded. We thus conclude that $E_m$ tends weakly to $E$ and, in particular, the continuity equation (6) is also satisfied in the limit. As $(v_t)_t$ is bounded on $Q_r$, uniformly in time, one has $\int_0^{t_r} \int_\Omega |v_t(u)|^2 \mathrm{d}\mu_t(u)\mathrm{d}t < \infty$ which proves that $(\mu_t)_t$ is absolutely continuous in $W_2$.

**Step (iv).**   So far, we have shown the convergence, up to a subsequence, to a Wasserstein gradient flow on $[0, t_r]$: it remains to show that $\lim_{r \to \infty} t_r = \infty$. Since $F(\mu_{m,0}) \to F(\mu_0)$ and all paths $(\mu_{t,m})_t$ decrease monotonically the value of $F$, everything lies in a sublevel of $R$, where $dR$ is bounded. It follows that a uniform bound on the velocity of the particles with linear growth in $r$ is available and, by Grönwall's inequality, we obtain that $\lim_{r \to \infty} t_r = \infty$, just as in the end of the proof of Proposition 2.5. The theorem follows by combining this result with the uniqueness stated in Proposition 2.5.

## C   Convergence to global minimizers

We give in this section a proof of Theorems 3.3 and 3.5. All results have two versions: one in the 2-homogeneous setting (Assumptions 3.2) and its counterpart in the partially 1-homogeneous setting (Assumptions 3.4). We have displayed in Figure 4 the level sets of functions with these homogeneity properties, in order to highlight the differences between these two cases. The proofs tend to be more straightforward in the 2-homogeneous setting and they can be read independently of the other case. This section is organized as follows:

- In Section C.1, we justify the global optimality conditions.
- We give in Section C.2 a criteria for Wasserstein gradient flows to escape from neighborhoods of non-optimal stationary points, and we also characterize measures that can be limits of Wasserstein gradient flows. These results are valid for arbitrary initializations.
- In Section C.3, we prove that the assumption on the support of the initialization made in Theorems 3.3 and 3.5 is preserved by Wasserstein gradient flows.
- All these facts combined lead to a proof of Theorems 3.3 and 3.5 in Section C.4.

It will be often the case in the statements and in the proofs that they involve the projection $h^i(\mu)$ of a probability measure $\mu \in \mathcal{P}(\Omega)$ (with $i = 1, 2$) (introduced in Section A.2) instead of $\mu$ itself. This is motivated by two facts: (i) this projected measure it generally the object of interest in the optimization problem as it clears the redundancy caused by homogeneity and (ii) the assumptions that the projection $h^i(\mu_t)$ of a Wasserstein gradient flow converges is more reasonable than the convergence in $W_2$ of the original gradient flow, where generally no compactness is available.

(a) Positively 1-homogeneous in the vertical variable.　　(b) Positively 2-homogeneous.

Figure 4: Level sets of some functions on $\Omega = \mathbb{R}^2$ with homogeneity properties. The derivative $F'(\mu)$ of $F$, seen as a continuous function on $\Omega$ "looks like" plot (a) in the partially 1-homogeneous case and like plot (b) in the 2-homogeneous case. Wasserstein gradient flows of $F$ are simply set of particles $\mu_t$ that "slide down" such landscapes following the direction $-\nabla F'(\mu_t)$ (the subtlety being that the landscape itself depends on $\mu_t$). Minimizers $\mu^*$ of $F$ over $\mathcal{M}_+(\Omega)$ are characterized by the fact that this function $F'(\mu^*)$ is nonnegative on $\Omega$ and vanishes on the support of $\mu^*$. By homogeneity, it is sufficient to study these functions restricted to a subspace (dotted lines) as we do in the proofs.

## C.1　Optimality conditions (proof of Proposition 3.1)

Let us first remark that, by a first order Taylor expansion of $R$, we have that for all $\mu, \sigma \in \mathcal{M}(\Omega)$ with $F(\mu), F(\sigma) < \infty$, it holds $\int |F'(\mu)| d\sigma < \infty$ and

$$\frac{d}{d\epsilon} F(\mu + \epsilon \sigma)|_{\epsilon=0} = \int_\Omega F'(\mu) d\sigma \quad \text{with} \quad F'(\mu) : u \mapsto \langle R'(\int \Phi d\mu), \Phi(u) \rangle + V(u).$$

Let $\mu, \nu \in \mathcal{M}_+(\Omega)$ be such that $F(\nu), F(\mu) < \infty$, consider $\sigma := \nu - \mu$ and its Lebesgue decomposition $\sigma = f\mu + \sigma^\perp$ where $f \in L^1(\mu)$, $\delta^\perp \in \mathcal{M}_+(\Omega)$ is singular to $\mu$ (see [10, Thm. 4.3.2]). Clearly, by the above first order formula, it is necessary to have $F'(\mu) \geq 0$ everywhere with equality $\mu$-a.e., for $\mu$ to be a minimizer. It is also sufficient since in this case we have, by convexity,

$$0 = \frac{d}{dt} F(\mu + t\sigma)|_{t=0} \leq \frac{d}{dt} \left( (1-t)F(\mu) + tF(\nu) \right)|_{t=0} = F(\nu) - F(\mu).$$

## C.2　A criteria to escape from non-optimal stationary points

We now give a criteria for Wasserstein gradient flows to escape from non-optimal stationary points. It is valid both in the finite-particle regime and in the many-particle limit. Such a result supports the idea that, even in the finite-particle case (i.e. classical gradient flows), the point of view using measures is natural.

### C.2.1　The $2$-homogeneous case

We start with the positively 2-homogeneous setting which is slightly simpler. We consider the operator $h^2 : \mathcal{M}_+(\mathbb{R}^d) \to \mathcal{M}_+(\mathbb{S}^{d-1})$ defined in Appendix A.2. To simplify notations, measures on the sphere $\mathbb{S}^{d-1}$ are interpreted hereafter as measures on $\mathbb{R}^d$ concentrated on the sphere.

**Proposition C.1** (Criteria to espace local minima). *Under Assumptions 3.2, let $\mu \in \mathcal{M}_+(\mathbb{R}^d)$ be such that $F'(\mu)$ is not nonnegative. There exists $\epsilon > 0$ and a set $A \subset \Omega$ such that if $(\mu_t)_t$ is a Wasserstein gradient flow of $F$ satisfying $\|h^2(\mu) - h^2(\mu_{t_0})\|_{\mathrm{BL}} < \epsilon$ for some $t_0 \geq 0$ and $\mu_{t_0}(A) > 0$ then there exists $t_1 > t_0$ such that $\|h^2(\mu) - h^2(\mu_{t_1})\|_{\mathrm{BL}} \geq \epsilon$.*

*Such a set is given by $A = \{r\theta \; ; \; r \in \,]0, \infty[ \text{ and } \theta \in K\}$ where $K$ is the $(-\eta)$-sublevel set of the restriction of $F'(\mu)$ to the unit sphere, for some $\eta > 0$ that can be chosen arbitrarily close to $0$.*

*Proof.* Let $g_\mu : \mathbb{S}^{d-1} \to \mathbb{R}$ be the restriction of $F'(\mu)$ to the unit sphere, and first assume that $0$ is in the range of $g_\mu$. Let $-\eta < 0$ be a negative regular value of $g_\mu$, which is guaranteed to exist

(arbitrarily close to 0) thanks to Assumption 3.2 and let $K \subset \mathbb{S}^{d-1}$ be the corresponding sublevel set. By the regular value theorem, its boundary $\partial K = g_\mu^{-1}(-\eta)$ is a differentiable orientable compact submanifold (of the sphere) of dimension $d - 2$ and is orthogonal to the gradient field of $g_\mu$. Also, $V$ is differentiable on a neighborhood of $\partial K$, by the regular value property. It holds $g_\mu(\theta) < -\eta$ for $\theta \in K$ and $\nabla g_\mu(\theta) \cdot \vec{n}_\theta < -\beta$ for all $\theta \in \partial K$, where $\vec{n}_\theta$ is the unit normal vector to $\partial K$ pointing outwards, for some $\beta > 0$. In the following lemma, we show that these properties of $K$ are also satisfied in a neighborhood of $\mu$. We denote by $\| \cdot \|_{C^1}$ the maximum of the supremum norm of a function and the supremum norm of its gradient.

**Lemma C.2.** *Let $\phi$ be the restriction of $\Phi$ to the sphere and let $\tilde{g}_\mu : \theta \in \mathbb{S}^{d-1} \mapsto \langle R'(\int \Phi d\mu), \phi(\theta) \rangle$. For all $C_0 > 0$, there exists $\alpha > 0$ such that for all $\mu, \nu \in \mathcal{M}(\Omega)$, such that $\|h^2(\mu)\|_{\mathrm{BL}}, \|h^2(\mu)\|_{\mathrm{BL}} < \eta$ it holds*

$$\|\tilde{g}_\mu - \tilde{g}_\nu\|_{C^1} \leq \alpha \|\phi\|_{C^1}^2 \cdot \|h^2(\mu) - h^2(\nu)\|_{\mathrm{BL}}.$$

*Proof.* Let us introduce $\alpha > 0$ the Lipschitz constant of $dR$ on the set $\{\int \Phi d\mu \ ; \ \mu \in \mathcal{P}_2(\mathbb{R}^d) \ ; \ h^2(\mu) < C_0\}$ which is bounded in $\mathcal{F}$. It holds

$$
\begin{aligned}
\|\tilde{g}_\mu - \tilde{g}_\nu\|_{C^1} &\leq \alpha \|\phi\|_{C^1} \cdot \|\textstyle\int \Phi d\mu - \int \Phi d\nu\| \\
&\leq \alpha \|\phi\|_{C^1} \cdot \|\textstyle\int \phi dh^2(\mu) - \int \phi dh^2(\nu)\| \\
&\leq \alpha \|\phi\|_{C^1} \cdot \sup_{f \in \mathcal{F}, \|f\| \leq 1} \textstyle\int \langle f, \phi \rangle d(h^2(\mu) - h^2(\nu)) \\
&\leq \alpha \|\phi\|_{C^1}^2 \cdot \|h^2(\mu) - h^2(\nu)\|_{\mathrm{BL}}.
\end{aligned}
$$

where the last bound is due to the fact that $u \mapsto \langle f, \phi(u) \rangle$ is $\|\phi\|_{C^1}^2$-Lipschitz and upper bounded in norm by $\|\phi\|_{C^1}$ whenever $f \in \mathcal{F}$ satisfies $\|f\| \leq 1$, and can be extended from the sphere $\mathbb{S}^{d-1}$ to $\mathbb{R}^d$ as a Lipschitz function with the same constant. $\qquad \square$

We now fix a large enough $C_0 > 0$ and consider measures $\nu$ such that $\|h^2(\nu)\|_{\mathrm{BL}} < C_0$. By posing $\epsilon = \min\{\eta, \beta\}/(4\alpha M^2)$ where $\alpha > 0$ is given by the previous lemma, if $\|h^2(\nu) - h^2(\mu)\| < \epsilon$, then $g_\nu$ is upper bounded by $-\eta/2$ on $K$ and $\nabla g_\nu(\theta) \cdot \vec{n}_\theta < -\beta/2$ for all $\theta \in \partial K$. Now let us consider a Wasserstein gradient flow $(\mu_t)_t$ of $F$ such that $\mu_0$ is concentrated on $B(0, r_0)$ for some $r_0 > 0$ and, posing $\epsilon$ as above, we assume that $\|h^2(\mu_0) - h^2(\mu)\|_{\mathrm{BL}} < \epsilon$. As long as this holds, the condition $\|h^2(\mu_t)\|_{\mathrm{BL}} < C_0$ for Lemma C.2 to apply also holds (if $C_0$ was chosen large enough in the first place). Let $t_1 > 0$ be the first time such that $\|h^2(\mu_{t_1}) - h^2(\mu)\|_{\mathrm{BL}} \geq \epsilon$, which might a priori be infinite.

Consider the flow $X$ of Lemma B.4. By construction of the set $K$, any path $t \mapsto u_t = X(t, u_0)$ with $u_0 \in \mathbb{R}_+ K$ remains in $\mathbb{R}_+ K$ for $t \leq t_1$. Moreover, by positive 2-homogeneity of $F'(\mu_t)$, the radial component of the velocity field is lower bounded by $r \cdot \eta$ so $|u_t| \geq |u_0| \exp(\eta t)$. In particular, for $0 \leq t < t_1$ and $\xi > 0$,

$$h^2(\mu_t)(K) \geq (\xi \exp(\eta t))^2 \cdot \mu_0(]\xi, \infty[ \times K).$$

It follows that as long as, for some $\xi > 0$, $\mu_0(]\xi, \infty[ \times K) > 0$, then $h^2(\mu_t)(K)$ grows exponentially fast: this implies that $\mu_t$ leaves the $\| \cdot \|_{\mathrm{BL}}$ ball (notice that all measures here are nonnegative). Hence $t_1$ is finite. Finally, if we had not assumed that 0 is in the range of $g_\nu$ in the first place, then we could simply take $K = \mathbb{S}^{d-1}$ and, by similar arguments, find that $|h^2(\mu_t)|(\mathbb{S}^{d-1})$ grows exponentially fast for $t < t_1$ if $\mu_0(\mathbb{R}^d \setminus \{0\}) > 0$. $\qquad \square$

We now give a general property of the stationary points.

**Lemma C.3.** *Under Assumptions 3.2, let $(\mu_t)_t$ be a Wasserstein gradient flow of $F$. If $h^2(\mu_t)$ converges weakly to $\nu \in \mathcal{M}_+(\mathbb{S}^{d-1})$, then $F'(\nu)$ vanishes $\nu$-a.e.*

*Proof.* We again consider the function $\tilde{g}_\mu : \theta \in \mathbb{S}^{d-1} \mapsto \langle R'(\int \Phi d\mu), \phi(\theta) \rangle$ for any measure $\mu \in \mathcal{M}_+(\mathbb{R}^d)$. At a point $\theta \in \mathbb{S}^{d-1}$, the velocity field $(v_t)_t$ associated to the gradient flow is obtained by applying the 2-Lipschitz map $(\mathrm{id} - \mathrm{proj}_{\partial V(\theta)})$ to the vector of radial component $2\tilde{g}_{\mu_t}(\theta)$ and tangential component $\nabla \tilde{g}_{\mu_t}(\theta)$. It follows, by Lemma C.2, that $v_{\mu_t}$ converges uniformly to $v_\nu$ on the sphere and as a consequence, $\int |v_t(u)|^2 dh^2(\mu_t)(u) \to \int |v_\nu(u)|^2 dh^2(\nu)(u)$. By recalling the energy

identity for gradient flows $F(\mu_{s_1}) - F(\mu_{s_2}) = \int_{s_1}^{s_2} |v_t(u)|^2 \mathrm{d}\mu_t \mathrm{d}t$ (see [3, Eq. (11.2.4)]) and this last term also equals $\int_{s_1}^{s_2} |v_t(u)|^2 \mathrm{d}h^2(\mu_t)\mathrm{d}t$ because the velocity field is positively 1-homogeneous. This necessarily implies, since $F$ is lower bounded on $\mathcal{P}_2(\Omega)$, that $|v_\nu(u)| = 0$ for $\nu$-a.e. $u \in \mathbb{R}^d$. In particular, looking at the radial component which is $2F'(\nu)$, this implies that $F'(\nu)$ vanishes $\nu$-a.e. $\qquad\square$

### C.2.2 The partially 1-homogeneous case

For the partially 1-homogeneous case, we consider the operator $h^1 : \mathcal{M}_+(\Omega) \to \mathcal{M}(\Theta)$ defined in Appendix A.2.

**Proposition C.4** (Criteria to espace local minima). *Under Assumptions 3.4, let $\mu \in \mathcal{M}(\Omega)$ be such that $F'(\mu)$ is not nonnegative. Then there exists $\epsilon > 0$ and a set $A \subset \Omega$ such that if $(\mu_t)_t$ is a Wasserstein gradient flow of $F$ satisfying $\|h^1(\mu) - h^1(\mu_{t_0})\|_{\mathrm{BL}} < \epsilon$ for some $t_0 \geq 0$ and $\mu_{t_0}(A) > 0$ then there exists $t_1 > t_0$ such that $\|h^1(\mu) - h^1(\mu_{t_1})\|_{\mathrm{BL}} \geq \epsilon$.*

*Such a set is given by $A = (\mathbb{R}_+ \times K^+) \cup (\mathbb{R}_- \times K^-)$ where $K^+$ (respectively $K^-$) is the $(-\eta)$-sublevel set of $\theta \mapsto F'(\mu)(1,\theta)$ (respectively of $\theta \mapsto F'(\mu)(-1,\theta)$) for some $\eta > 0$ that can be chosen arbitrarily close to $0$.*

*Proof.* Let us suppose that $F'(\mu)$ takes a negative value on $\mathbb{R}_+ \times \Theta$ (the case where it takes its negative values only on $\mathbb{R}_- \times \Theta$ is similar) and let us introduce $g_\mu : \Theta \to \mathbb{R}$ the restriction of $F'(\mu)$ to $\{1\} \times \Theta$, that is $g_\mu(\theta) = \langle R'(\int \Phi \mathrm{d}\mu), \phi(\theta)\rangle + \tilde{V}(\theta)$. Let $-\eta < 0$ be a negative regular value of $g$, which is guaranteed to exist (arbitrarily close to 0) thanks to Assumption 3.4, and let $K^+ \subset \Theta$ be the corresponding sublevel set. By the regular value theorem, its boundary $\partial K^+ = g_\mu^{-1}(-\eta)$ is a differentiable orientable manifold of dimension $d - 2$ and is orthogonal to the gradient field of $g_\mu$. In the case where $\Theta$ is bounded, $\partial K^+$ is compact and, as a consequence, there is $\beta > 0$ such that $\inf_{\theta \in \partial K} |dg_\mu(\theta)| \geq \beta$. If $\Theta = \mathbb{R}^{d-1}$ and the sublevel set $K^+$ is unbounded, then we have to choose $\eta$ so that it is also a regular value of the function on the sphere $\mathbb{S}^{d-2}$ to which $g_\mu$ converges uniformly at infinity. Then, the same positive lower bound holds for some $\beta > 0$. It follows that on $K$, $g_\mu \leq -\eta$ and, $\nabla g_\mu(\theta) \cdot \vec{n}_\theta < -\beta$ for all $\theta \in \partial K$, where $\vec{n}_\theta$ is the unit normal vector to $\partial K$ pointing outwards. In the following lemma, we show that these properties of $K$ are also true with respect to $g_\nu$ if $\nu$ is close enough to $\mu$. We denote by $\|\cdot\|_{C^1}$ the maximum of the supremum norm of a function and the supremum norm of its gradient.

**Lemma C.5.** *For all $C_0 > 0$, there exists $\alpha > 0$ such that for all $\mu, \nu \in \mathcal{M}_+(\Omega)$ that satisfy $\|h^1(\mu)\|_{\mathrm{BL}}, \|h^1(\nu)\|_{\mathrm{BL}} < C_0$, it holds*

$$\|g_\mu - g_\nu\|_{C^1} \leq \alpha \|\phi\|_{C^1}^2 \cdot \|h^1(\mu) - h^1(\nu)\|_{\mathrm{BL}}.$$

*Proof.* Let us introduce $\alpha > 0$ the Lipschitz constant of $dR$ on the set $\{\int \Phi \mathrm{d}\mu \ ; \ \mu \in \mathcal{P}(\mathbb{R}^d) \ ; \ h^1(\mu) < C_0\}$ which is bounded in $\mathcal{F}$. It holds

$$
\begin{aligned}
\|g_\mu - g_\nu\|_{C^1} &\leq \alpha \|\phi\|_{C^1} \cdot \|\textstyle\int \Phi \mathrm{d}\mu - \int \Phi \mathrm{d}\nu\| \\
&\leq \alpha \|\phi\|_{C^1} \cdot \|\textstyle\int \phi \mathrm{d}h^1(\mu) - \int \phi \mathrm{d}h^1(\nu)\| \\
&\leq \alpha \|\phi\|_{C^1} \cdot \sup_{f \in \mathcal{F}, \|f\| \leq 1} \textstyle\int \langle f, \phi\rangle \mathrm{d}(h^1(\mu) - h^1(\nu)) \\
&\leq \alpha \|\phi\|_{C^1}^2 \cdot \|h^1(\mu) - h^1(\nu)\|_{\mathrm{BL}}.
\end{aligned}
$$

where the last bound is due to the fact that $u \mapsto \langle f, \phi(u)\rangle$ is $\|\phi\|_{C^1}$-Lipschitz and upper bounded in norm by $\|\phi\|_{C^1}$ whenever $f \in \mathcal{F}$ satisfies $\|f\| \leq 1$. $\qquad\square$

We now fix a large enough $C_0 > 0$ and consider measures $\nu$ such that $\|h^1(\nu)\|_{\mathrm{BL}} < C_0$. By posing $\epsilon = \min\{\eta, \beta\}/(4\alpha M^2)$ where $\alpha > 0$ is given by the previous lemma, if $\|h^1(\nu) - h^1(\mu)\| < \epsilon$, then $g_\nu$ is upper bounded by $-\eta/2$ on $K$ and $\nabla g_\nu(\theta) \cdot \vec{n}_\theta < -\beta/2$ for all $\theta \in \partial K$. Now, let us consider a Wasserstein gradient flow $(\mu_t)_t$ of $F$ such that $\mu_0$ is concentrated on $[-r_0, r_0] \times \Theta$ for some $r_0 > 0$ and $\|h^1(\mu_0) - h^1(\mu)\|_{\mathrm{BL}} < \epsilon$. As long as this holds, the condition $\|h^1(\mu_t)\|_{\mathrm{BL}} < C_0$ for Lemma C.5 to apply also holds. Let $t_1 > 0$ be the first time such that this last condition is not satisfied, which might a priori be infinite.

Consider the flow $X$ of Lemma B.4. By construction of the set $K^+$, any path $t \mapsto (w_t, \theta_t) = X(t, (w_0, \theta_0))$ with $(w_0, \theta_0) \in \mathbb{R}_+ \times K$ remains in $\mathbb{R}_+ \times K^+$ for $t \leq t_1$. Moreover, by homogeneity of $F'(\mu_t)$ in the variable $w$, the component of the velocity field on $w$ is lower bounded by $\eta/2$ so $w_t \geq w_0 + t \cdot \eta/2$. For similar reasons, no path enters the set $\mathbb{R}_- \times K^+$ during this time interval and the paths inside this set satisfy $w_t \geq w_0 + t \cdot \eta/2$ (this follows by the fact that $F'(-1, \cdot) \geq F'(1, \cdot)$). In particular, for $0 \leq t < t_1$,

$$h^1(\mu_t)(K^+) \geq (t \cdot \eta/2) \cdot \mu_0(\mathbb{R}_+ \times K^+) + \min\{0, t \cdot \eta/2 - r_0\} \cdot \mu_0(\mathbb{R}_- \times K^+).$$

So we see that as long as $\mu_0(\mathbb{R}_+ \times K) > 0$, then $h^1(\mu_t)(K^+)$ grows at least linearly.

If $\Theta = K^+$ then the previous lower bound immediately implies that $t_1$ is finite (choose the constant unit function in the definition of the norm $\|\cdot\|_{\mathrm{BL}}$). Otherwise, in order to finalize our proof, we need to make sure that the mass $h^1(\mu_t)(K^+)$ does not grow unbounded just near the boundary of $K^+$. To do so, let us consider another sublevel set $\tilde{K}^+$ of $g_\mu$ associated to another regular value in the range $\tilde{\eta} \in ]-\eta, 0[$ and such that $\tilde{K}^+$ does not cover $\Theta$. As $g_\mu$ is Lipschitz, there exists $\Delta \in ]0, 1]$ such that the distance between $K^+$ and $\Theta \setminus \tilde{K}^+$ is at least $\Delta$. Taking another, smaller radius $\epsilon > 0$ if necessary, by similar arguments as above, either $t_1$ is smaller than $2r_0/\tilde{\eta}$, or there exists $\tilde{t} > t_0$ such that $h^1(\mu_t)$ is nonnegative on $\tilde{K}^+$ for $t \in [\tilde{t}, t_1[$. Choosing, as a test function in the definition of the norm $\|\cdot\|_{\mathrm{BL}}$, the distance to the set $\Theta \setminus \tilde{K}$ clipped to 1, one obtains, for $t \in [\tilde{t}, t_1[$,

$$\|h^1(\mu_t)\|_{\mathrm{BL}} \geq \Delta \cdot h^1(\mu_t)(K^+)$$

which also grows at least linearly with $t$. So $h^1(\mu_t)$ eventually leaves any $\|\cdot\|_{\mathrm{BL}}$-ball, hence $t_1$ is finite. $\qquad\square$

As for the 2-homogeneous case, we give a general property of the stationary points.

**Lemma C.6.** *Under Assumptions 3.4, let $(\mu_t)_t$ be a Wasserstein gradient flow of $F$. If $h^1(\mu_t)$ converges weakly to $\nu \in \mathcal{M}_+(\Theta)$, then $F'(\nu)$ vanishes $\nu$-a.e.*

*Proof.* At a point $(1, \theta) \in \Omega$, with $\theta \in \Theta$, the velocity field $(v_t)_t$ associated to the gradient flow is given by applying the 2-Lipschitz map $(\mathrm{id} - \mathrm{proj}_{\partial V(1, \theta)})$ to the vector with first component $g_{\mu_t}(\theta)$ and other components $\nabla g_{\mu_t}(\theta)$, where $g_{\mu_t}$ is defined as in the proof of Proposition C.4. It follows, by Lemma C.2, that $v_{\mu_t}$ converges uniformly to $v_\nu$ on $\{1\} \times \Theta$. However, in contrast to Lemma C.6, the energy dissipation identity is not invariant by the projection operator, so we have to develop arguments similar to those used to prove Proposition C.4 (we do so with less details). Using the uniform convergence of $g_{\mu_t}$ if there exists $\theta_0 \in \Theta$ such that $g_\nu(\theta) > 0$ then we can build a set $\mathbb{R}^+ \times K$ with $\theta_0 \in \mathrm{int} K$ such that for some $t_0 > 0$, no trajectory of the flow $X_t$ enters this set after $t_0$ and the component of the velocity on $w$ is upper bounded by $-g_\nu(\theta)/2$. Since $\mu_{t_0}$ is concentrated on a set $Q_{r_0}$, this implies that $\mu_{t_0}(\mathbb{R}_+^* \times K)$ vanishes in finite time and in particular, $\nu(K) = 0$. Thus we have shown that $F'(\nu)$ is nonpositive $\nu$-a.e. Also it can be deduced by Proposition C.4, that $F'(\nu)$ is nonnegative $\nu$-a.e. So $F'(\nu)$ vanishes $\nu$-a.e. $\qquad\square$

### C.3 Stability of separation properties

Here we prove the fact that the separation properties of the support used in Theorems 3.5 and 3.3 are preserved by Wasserstein gradient flows. We give a proof based on *topological degree* theory: this tool allows to cover the case of discontinuous velocity fields, which appear when $V$ is non-differentiable. In a more regular setting, the facts that follow are easier to prove because then, $\mu_t$ is the pushforward of $\mu_0$ by a homeomorphism. Let us give a definition of the topological degree sufficient to our setting.

**Definition C.7** (Topological degree). *Let $f : \mathbb{R}^d \to \mathbb{R}^d$ be a continuous map, $A \subset \mathbb{R}^d$ a bounded open set and $y \notin f(\partial A)$. The topological degree $\deg(f, A, y)$ is a signed integer that satisfies:*

1. *If $\deg(f, A, y) \neq 0$ then there exists $x \in A$ such that $f(x) = y$. If $y \in A$ then $\deg(\mathrm{id}, A, y) = 1$.*

2. *If $A_1, A_2$ are disjoint open subsets of $A$ and $y \notin f(\overline{A} \setminus (A_1 \cup A_2))$ then $\deg(f, A, y) = \deg(f, A_1, y) + \deg(f, A_2, y)$.*

3. *If $X : [0, 1] \times \mathbb{R}^d \to \mathbb{R}^d$ is continuous and $y : [0, 1] \to \mathbb{R}^d$ is a continuous curve such that $y(t) \notin X_t(\partial A)$ for all $t \in [0, 1]$, then $\deg(X_t, A, y_t)$ is constant on $[0, 1]$.*

These properties characterize a uniquely well-defined map $\deg$ from the set of triplets $(f, A, y)$ as above to the set of signed integers [8, Thm. 1-2]. Intuitively, it gives an *algebraic* count of the number of solutions to $f(x) = y$ for $x \in A$, where algebraic means that a solution $x$ counts as $+1$ if $f$ preserves orientation around $x$ and $-1$ otherwise.

The following lemma shows that taking the support of a measure and its pushforward by a continuous map are operations that almost commute. They commute for instance if the map is *closed* (i.e. maps closed sets to closed set).

**Lemma C.8.** *If $f : \mathbb{R}^d \to \mathbb{R}^d$ is a continuous map and $\mu \in \mathcal{M}_+(\mathbb{R}^d)$, then $\mathrm{spt}(f_\# \mu) = \overline{f(\mathrm{spt}\,\mu)}$.*

*Proof.* Let $y \in f(\mathrm{spt}\,\mu)$ and $\mathcal{V}$ a neighborhood of $y$. By continuity, $f^{-1}(\mathcal{V})$ is the neighborhood of a point in $\mathrm{spt}\,\mu$ so $0 < \mu(f^{-1}(\mathcal{V})) = f_\# \mu(\mathcal{V})$, hence $y \in \mathrm{spt}\,f_\# \mu$ so $f(\mathrm{spt}\,\mu) \subset \mathrm{spt}\,f_\# \mu$. Conversely, let $y \in \overline{f(\mathrm{spt}\,\mu)}^c$ and let $\mathcal{V}$ a neighborhood of $y$ that does not intersect $\overline{f(\mathrm{spt}\,\mu)}$. This neighborhood satisfies $f^{-1}(\mathcal{V}) \subset (\mathrm{spt}\,\mu)^c$, so it holds $f_\# \mu(\mathcal{V}) = \mu(f^{-1}(\mathcal{V})) \leq \mu((\mathrm{spt}\,\mu)^c) = 0$. Hence $y \in (\mathrm{spt}\,f_\# \mu)^c$ so $\overline{f(\mathrm{spt}\,\mu)}^c \subset (\mathrm{spt}\,f_\# \mu)^c$ which implies $\mathrm{spt}\,f_\# \mu \subset \overline{f(\mathrm{spt}\,\mu)}$. $\square$

### C.3.1 The 2-homogeneous case

We first state the property and the stability result that we wish to establish in the 2-homogeneous setting.

**Property C.9** (Separation, 2-homogeneous case). *$K$ is a closed subset of $\mathbb{R}^d$ contained in $B(0, r_b)$ that separates $r_a \mathbb{S}^{d-1}$ from $r_b \mathbb{S}^{d-1}$, for some $0 < r_a < r_b$.*

**Lemma C.10** (Stability of the separation property). *Under Assumptions 3.2, let $(\mu_t)_{t \geq 0}$ be a Wasserstein gradient flow of $F$. If the support of $\mu_0$ satisfies Property C.9, so does the support of $\mu_t$, for all $t > 0$.*

Note that this property is generally lost *in the limit $t \to \infty$*. This lemma is a consequence of the following, more abstract proposition, that deals with sets instead of measures. The reader can keep in mind that we will apply this result with $X$ being the flow of the velocity field introduced in Lemma B.4 and $K$ being the support of $\mu_0$.

**Proposition C.11** (Set separation, spheres). *Consider a continuous map $X : [0, T] \times \mathbb{R}^d \to \mathbb{R}^d$ such that $X(0, \cdot) = \mathrm{id}$, and such that, for all $\epsilon > 0$, there exists $\eta > 0$ such that $u \in B(0, \eta)$ implies $X_t^{-1}(u) \subset B(0, \epsilon)$. If $K$ satisfies Property C.9, then $X_t(K)$ satisfies the same property for all $t \in [0, T]$.*

*Proof.* Let $0 < \epsilon < \alpha < \beta$ be such that $X_t(K) \subset B(0, \alpha - \epsilon)$ and $B(0, \alpha + \epsilon) \subset X_t(B(0, \beta))$ for all $t \in [0, T]$ and let $A$ be the intersection of $B(0, \beta)$ with the (unique) unbounded connected component of $\mathbb{R}^d \setminus K$. Consider the function $\tilde{X} : (t, x) \mapsto (t, X_t(x))$ and the set $S = \tilde{X}([0, T] \times \partial A)$ which is a compact subset of $[0, T] \times \mathbb{R}^d$. Since connected components of $S^c$ (the complement of $S$ in $[0, T] \times \mathbb{R}^d$) are path connected, recalling Definition C.7, it follows that

$$(t, x) \mapsto \deg(X_t, A, x)$$

is constant on each connected component of $S^c$. Moreover, this degree equals 1 if the connected component intersects $\{0\} \times A$ and 0 if it intersects $\{0\} \times (\mathbb{R}^d \setminus A)$. In particular, this degree is 1 on $[0, T] \times \alpha \mathbb{S}$ and, by the assumptions on $K$ and $X$, there is a small tube $[0, T] \times B(0, \eta)$ where the above degree is 0. So for a fixed $t \in [0, T]$, any path joining $(\eta/2) \mathbb{S}^{d-1}$ to $\alpha \mathbb{S}^{d-1}$ must intersect $X_t(\partial A)$. We restrict our attention to paths confined in $B(0, \alpha)$ and it remains to notice that $\partial A \subset K \cup \beta \mathbb{S}$ and so

$$X_t(\partial A) \cap B(0, \alpha) \subset (X_t(K) \cup X_t(\beta \mathbb{S})) \cap B(0, \alpha) = X_t(K)$$

This shows that any path joining $(\eta/2) \mathbb{S}$ to $\alpha \mathbb{S}$ must intersect $X_t(K)$. $\square$

*Proof of Lemma C.10.* Consider the continuous flow $X : [0, T] \times \mathbb{R}^d \to \mathbb{R}^d$ introduced in Lemma B.4. For all $t \in [0, T]$, the map $X_t$ is coercive and thus closed, so Lemma C.8 applies and gives $\mathrm{spt}((X_t)_\# \mu_0) = X_t(\mathrm{spt}\,\mu_0)$. We just have to check the assumption of Proposition C.11 concerning the stability of the inverse map of $X_t$ near 0. Since $\Phi$ and $V$ are 2-homogeneous, we have that $d\Phi$ and $\partial V$ that are 1-homogeneous and thus, there exists a constant $C > 0$ such that $|v_t(u)| \leq C|u|$ for all $t \in [0, T]$ and $u \in \mathbb{R}^d$. This upper bound on the velocity implies in particular that if $u_0$ is at a distance $|u_0|$ from 0, then it is at least at a distance $|u_0| \exp(-Ct)$ for $t \in [0, T]$. $\square$

### C.3.2 The partially 1-homogeneous case

Here are the analogous separation property and stability lemma for the partially 1-homogeneous case.

**Property C.12** (Separation, partially 1-homogeneous case). *$K$ is a closed set contained in a box $Q_r := [-r, r] \times \Theta$ and separates $\{-r\} \times \Theta$ from $\{r\} \times \Theta$ for some $r > 0$ (in the ambient space $\Omega = \mathbb{R} \times \Theta$).*

**Lemma C.13** (Stability of the separation property). *Under Assumptions 3.4, let $(\mu_t)_{t \geq 0}$ be a Wasserstein gradient flow of $F$. If the support of $\mu_0$ satisfies Property C.12, then so does the support of $\mu_t$, for all $t > 0$.*

Similarly as above, we first prove an abstract topological result.

**Proposition C.14** (Set separation, boxes). *Let $\Theta \subset \mathbb{R}^d$ be the closure of a bounded, connected, open set and, for some $T > 0$, let $X : [0, T] \times (\mathbb{R} \times \Theta) \to \mathbb{R} \times \Theta$ be a continuous map such that $X(0, \cdot) = \mathrm{id}$ and $X_t(\mathbb{R} \times \partial\Theta) \subset \mathbb{R} \times \partial\Theta$ for all $t \in [0, T]$. If $K$ satisfies Property C.12, then $X_t(K)$ satisfies Property C.12 for all $t \in [0, T]$.*

*Proof.* Let $0 < \epsilon < \alpha < \beta$ be such that $X_t(K) \subset \,]-\alpha - \epsilon, \alpha + \epsilon[\, \times \Theta$ and $[-\alpha, \alpha] \times \Theta \subset X_t(\,]-\beta - \epsilon, \beta + \epsilon[\, \times \Theta)$ for all $t \in [0, T]$, and let $A$ be the intersection of $\,]-\beta, \beta[\times\Theta$ with the (unique) connected component of $(\mathbb{R} \times \Theta) \setminus K$ that contains $\{\alpha\} \times \Theta$. The set $A$ is bounded and open in $\mathbb{R} \times \mathbb{R}^{d-1}$. Consider the function $\tilde{X} : (t, x) \mapsto (t, X_t(x))$ and the set $S = \tilde{X}([0, T] \times \partial A)$ which is a compact subset of $[0, T] \times (\mathbb{R} \times \Theta)$. Since connected components of $S^c$ (the complement of $S$ in $[0, T] \times (\mathbb{R} \times \Theta)$) are path connected, recalling Definition C.7, it follows that

$$(t, (w, \theta)) \mapsto \deg(X_t, A, (w, \theta))$$

is constant on each connected component of $S^c$. Moreover, this degree is 1 on $[0, T] \times (\{\alpha\} \times \Theta)$ and is 0 on $[0, T] \times (\{-\alpha\} \times \Theta)$. So for a fixed $t \in [0, T]$, any path joining $\{-\alpha\} \times \Theta$ to $\{\alpha\} \times \Theta$ must intersect $X_t(\partial A)$. It is in particular true for paths entirely contained in $[-\alpha, \alpha] \times \mathrm{int}\,\Theta$. It remains to notice that $\partial A \subset K \cup (\mathbb{R} \times \partial\Theta) \cup (\{\beta\} \times \Theta)$ and so, thanks to our assumption on $X$,

$$X_t(\partial A) \cap ([-\alpha, \alpha] \times \mathrm{int}\,\Theta) \subset X_t(K).$$

This shows that $X_t(K)$ separates $\{-\alpha\} \times \Theta$ from $\{\alpha\} \times \Theta$ in $\mathbb{R} \times \mathrm{int}\,\Theta$ and in fact also in $\mathbb{R} \times \Theta$ because $X_t(K)$ is closed. $\qquad\square$

*Proof of Lemma C.13.* Let $X$ be the flow of the velocity fields introduced in Lemma B.4. It is continuous and satisfies $\mu_t = (X_t)_{\#}\mu_0$. Moreover, $X_0 = \mathrm{id}$ and $X_t$ is closed because it is coercive. We have to deal with two cases: when $\Theta$ is bounded and when $\Theta = \mathbb{R}^{d-1}$. In the first case, by Lemma C.8, it is sufficient so verify the assumptions of Proposition C.14. This reduces to making sure that $X_t(\mathbb{R} \times \partial\Omega) \subset \mathbb{R} \times \partial\Omega$, which is guaranteed by the Neumann boundary conditions. For the unbounded case, we bring ourselves back to the bounded case, by means of the diffeomorphism $\psi : \mathbb{R} \times \mathbb{R}^d \to \mathbb{R} \times \mathrm{int}\, B(0, 1)$ defined by $\psi(w, \theta) = (w, (\theta/|\theta|) \cdot \tanh|\theta|)$ if $\theta \neq 0$ and $\psi(w, 0) = (w, 0)$ otherwise. Let $Y_t := \Psi \circ X_t \circ \Psi^{-1}$ be the flow where the second variable is mapped to the open unit ball. By direct calculus, one sees that $Y_t$ is the flow of the velocity field $\tilde{v}_t(y) = d\psi_{\psi^{-1}(y)}(v_t \circ \psi^{-1}(y))$ defined on $\mathbb{R} \times \mathrm{int}\, B(0, 1)$ and that can be extended by continuity on $\mathbb{R} \times \mathbb{S}^{d-2}$ by $(g_\infty(\theta) \cdot \mathrm{sign}\, w, 0)$ where $g_\infty$ is the limit which existence is assumed in Assumption 3.4-(iii) and, here, $\mathrm{sign}\, 0 = 0$. As $Y_t$ satisfies the properties of Proposition C.14, the conclusion of this proposition holds for the set $\psi(\mathrm{spt}\,\mu_t) = \psi \circ X_t(\mathrm{spt}\,\mu_0)$. Since $\psi$ is a diffeomorphism, it preserves connectedness properties and Lemma C.13 is proved. $\qquad\square$

### C.4 Main theorems: proofs and generalization

First, let us state a lemma that relates the convergence of the Wasserstein gradient flows to an *asymptotic property* for the classical gradient flows, when $m, t \to \infty$. This result is used in the last claims of Theorems 3.3 and 3.5.

**Lemma C.15.** *Under Assumptions 2.1, let $(\mu_t)$ be a Wasserstein gradient flow which initialization is concentrated on a set $Q_{r_0}$ and such that $F(\mu_t) \to F^*$. If $(\mu_{0,m})_m$ is a sequence of measures concentrated on a set $Q_{r_0}$ that converges to $\mu_0$ in $W_2$, then*

$$F^* = \lim_{t \to \infty} \lim_{m \to \infty} F(\mu_{m,t}) = \lim_{m \to \infty} \lim_{t \to \infty} F(\mu_{m,t}).$$

*Proof.* The first double limit where $m$ goes first to $\infty$ is a consequence of Theorem 2.6 combined with the continuity of $F$ for the Wasserstein metric, proved in Lemma A.1. The other double limit is obtained by the mononicity of $F(\mu_t)$ along Wasserstein gradient flows. Indeed, for all $\epsilon > 0$, there exists $t_0 \in \mathbb{R}_+$ such that $F(\mu_{t_0}) < F^* + \epsilon/2$ and by Theorem 2.6, there is $m_0 \in \mathbb{N}$ such that for all $m \geq m_0$, $F(\mu_{t_0,m}) < F(\mu_{t_0}) + \epsilon/2$. Since $t \mapsto F(\mu_{m,t})$ is decreasing and lower bounded for all $m \in \mathbb{N}$, it follows

$$\forall m > m_0, \ \lim_{t \to \infty} F(\mu_{m,t}) \leq F(\mu_{m,t_0}) < F^* + \epsilon$$

which proves the second limit. $\qquad\square$

### C.4.1 The 2-homogeneous case

**Theorem C.16.** *Under the assumptions of Theorem 3.3, if $h^2(\mu_t)$ converges weakly, then its limit is a global minimizer of $F$ over $\mathcal{M}_+(\Omega)$ and $\lim_{t \to \infty} F(\mu_t) = F^*$.*

This statement is stronger than Theorem 3.3: indeed, if $\mu_t$ converges for the Wasserstein metric, then $h^2(\mu_t)$ converges weakly (but the converse is generally not true).

*Proof.* Let $\nu \in \mathcal{M}_+(\mathbb{S}^{d-1})$ be the weak limit of $h^2(\mu_t)$. By Lemma C.3, $F'(\nu)$ vanishes $\nu$-a.e. For the sake of contradiction, assume that $\nu$ is not a minimizer of $F$ over $\mathcal{M}_+(\Omega)$: this implies that $F'(\nu)$ is not nonnegative. Let $A \subset \Omega$ and $B_{\mathrm{BL}} \subset \mathcal{M}(\mathbb{S}^{d-1})$ be the set and the $\|\cdot\|_{\mathrm{BL}}$-ball provided by Proposition C.1. As $h^2(\mu_t)$ converges weakly, there exists $t_0 > 0$ such that for all $t > t_0$, $h^2(\mu_t) \in B_{\mathrm{BL}}$. But by Lemma C.10, $\mu_{t_0}(A) > 0$ and, by Proposition C.1, there exists $t_1 > t_0$ such that $\mu_{t_1} \notin B_{\mathrm{BL}}$, which is a contradiction so $\nu$ is minimizer of $F$ over $\mathcal{M}_+(\Omega)$. The second claim is a consequence of the continuity of $F$ (Lemma A.1). $\qquad\square$

### C.4.2 The partially 1-homogeneous case

Again, we prove a statement in terms of the projected measures: Theorem 3.5 can be deduced as an immediate corollary. Some highlights of the proof are given in Figure 5.

**Theorem C.17.** *Under the assumptions of Theorem 3.5, if $h^1(\mu_t)$ converges weakly, then its limit is a global minimizer of $F$ over $\mathcal{M}_+(\Omega)$ and $\lim_{t \to \infty} F(\mu_t) = F^*$.*

*Proof.* Let $\nu$ be the weak limit of $h^1(\mu_t)$ and we see it as a measure on $\{1\} \times \Theta$. By Lemma C.6, $F'(\nu)$ vanishes $\nu$-a.e. For the sake of contradiction, assume that $\nu$ is not a minimizer of $F$ over $\mathcal{M}_+(\Omega)$: this implies that $F'(\nu)$ is not nonnegative. Let $A \subset \Omega$ and $\epsilon$ be the set and the radius of the $\|\cdot\|_{\mathrm{BL}}$-ball which are provided by Proposition C.1. As $h^1(\mu_t)$ converges weakly, there exists $t_0 > 0$ such that for all $t > t_0$, $\|h^1(\mu_t) - \nu\|_{\mathrm{BL}} < \epsilon$. In the favorable case where $\mu_{t_0}(A) > 0$ then we can conclude as in the 2-homogeneous case, *but this is not immediately guaranteed by Lemma C.10*: the situation is thus trickier than in the proof of the 2-homogeneous case.

We take notations from the proof of Proposition C.4 and consider first the case when $\Theta$ is bounded. Let $\theta_0 \in K^+$ be a local minimum of $g_\nu$ in the interior of $K^+$ relatively to $\Theta$ (the case when $K^+$ is empty but $K^-$ is not could be treated similarly). Thanks to Neumann boundary conditions, it holds $\nabla g_\nu(\theta_0) = 0$, even when $\theta_0$ lies on the boundary of $\Theta$. By Lemma C.10, the line $\mathbb{R} \times \{\theta_0\}$ intersects the support of $\mu_{t_0}$. If this intersection lies in $\mathbb{R}_+ \times K^+$, we can conclude immediately by Proposition C.1. Otherwise, we fix $M > 0$ such that $\mu_{t_0}$ is concentrated on $[-M, M] \times \Theta$ and we resort to applying Lemma C.18 below.

Let $r_0 > 0$ be such that $B(\theta_0, r_0) \cap \Theta \subset K^+$. By Lemma C.18, there exists $t_1 > t_0$ such that if the support of $\mu_{t_1}$ intersects $[-M, 0] \times \{\theta_0\}$ then it intersects $\mathbb{R}_+ \times K^+$ at a subsequent time and again, we can conclude by Proposition C.1. So it remains to check that the support of $\mu_{t_1}$ intersects $[-M, 0] \times \{\theta_0\}$; the difficulty here is that $M$ was chosen prior to $t_1$. The justification is as follows: by Lemma C.10, the support of $\mu_{t_1}$ intersects $\mathbb{R}_- \times \{\theta_0\}$ at a point $(w_0, \theta_0)$. The properties of $K^+$ imply that the pre-image by the flow $X_t$ of $(w_0, \theta_0)$ is included in $[-M, 0[ \times K^+$ and, since the $w$-component of the velocity field is lower bounded on $\mathbb{R}_- \times K^+$ by $\eta/2$ for $t > t_0$, one has $w_0 > M$.

As for the case when $\Theta = \mathbb{R}^{d-1}$, we can reproduce the proof above by mapping the flow to the unit sphere, as done in the proof of Lemma C.13. The last claim of the theorem is a consequence of the continuity of $F$ (Lemma A.1). $\qquad\square$

Figure 5: Example, in the partially 1-homogeneous setting, of a stationary point $\nu$ that is not a minimizer of $F$ over $\mathcal{M}_+(\Omega)$ since $F'(\nu)$ is not nonnegative. The variable $w$ corresponds to the vertical axis, the support of $\nu$ is the discrete set of red dots, and the lines show the level sets of $F'(\nu)$. There is a neighborhood of $\nu$ such that if a Wasserstein gradient flow $\mu_t$ enters this neighborhood and gives mass to a certain set (shown in red) then $\mu_t$ subsequently escapes this neighborhood. In Proposition C.4 this is proved for the lower part of this set (under the horizontal axis, where $F'(\nu)$ is negative). Technical Lemma C.18 is concerned with building the upper part of this set. The proof of Theorem C.17 lies on the fact that any measure satisfying the separation Property C.12 gives mass to this set.

In the unfavorable case encountered in the proof of Theorem C.17, we had to invoke the following lemma. It has a different nature than the other results of this paper because it relies on an explicit integration of the trajectories of the gradient flow, which means that it depends on the choice of the metric.

**Lemma C.18.** *Consider, for a measure $\nu \in \mathcal{M}(\Theta)$, a point $\theta_0 \in \Theta$ such that $|\nabla g_\nu(\theta)| = 0$ and $g_\nu(\theta) \leq -\eta$ for some $\eta > 0$. For any $M > 0$ and $r_0 > 0$, there exists $T, \epsilon > 0$ such that if $(\mu_t)_t$ is a Wasserstein gradient flow of $F$ that satisfies for all $t \in [0, T]$, $\|g_{\mu_t} - g_\nu\|_{C^1} \leq \epsilon$ and denoting $(w(t), \theta(t))$ the solution of the flow of Lemma B.4 starting from $(w_0, \theta_0)$ with $w_0 \in [-M, 0]$, it holds $w(T) = 0$ and $|\theta(T) - \theta_0| < r_0$.*

*Proof.* The Lipschitz regularity of $g_\nu$ and its derivative implies that there exists $L > 0$ such that $\max\{|g_\nu(\theta) - g_\nu(\theta_0)|, |\nabla g_\nu(\theta) - \nabla g_\nu(\theta_0)|\} \leq L|\theta - \theta_0|$ for all $\theta \in \Theta$. Without loss of generality, we assume that $r_0 < \eta/(4L)$. Consider $\epsilon \in ]0, \eta/4[$ and assume that there exists $\bar{T} > 0$ such that $\|g_{\mu_t} - g_\nu\|_{C^1} \leq \epsilon$ for $t \in [0, \bar{T}]$. Writing $q(t) = |\theta(t) - \theta_0|$, it holds for $t \in [0, \bar{T}]$,

$$
\begin{cases}
\dfrac{dq}{dt} \leq -w(\epsilon + Lq) \\[2mm]
\dfrac{dw}{dt} \geq \eta - \epsilon - Lq
\end{cases}
$$

In particular, if we can make sure that $|q(t)| < \bar{r}$ for $t \in [0, \bar{T}]$ and if $\bar{T} > 2/\eta$ then, as $(dw/dt) \geq \eta/2$ on this interval, there exists $T < 2/\eta$ such that $w(T) = 0$.

It remains to make sure that we indeed have $|q(t)| < \bar{r}$ for $t \in [0, T]$, by adjusting if necessary the value of $\epsilon$. Parametrizing in $w$ instead of $t$ (it is an admissible reparametrization thanks to the positive lower bound on its derivative), we get

$$
dq/dw = (dq/dt) \cdot (dt/dw) \leq -w(\epsilon + Lq) \cdot 2/\eta.
$$

We can apply Grönwall's lemma to $\tilde{q}(w) = \epsilon + Lq(w)$ which satisfies $(d/dw)\tilde{q}(w) \leq (-2L/\eta) \cdot w \cdot \tilde{q}(w)$ and obtain

$$
\tilde{q}(w) \leq \tilde{q}(w_0) \exp\left(-(2L/\eta) \int_{w_0}^{0} s\, ds\right) = \epsilon \exp(Lw_0^2/\eta).
$$

Thus, choosing $\epsilon < Lr_0/(\exp(Lw_0^2/\eta) - 1)$, it is guaranteed that $q(t) \leq r_0$ for $t \in [0, T]$. $\qquad\square$

## C.5 Remarks

We conclude this theoretical section with two opening remarks related to the global convergence theorems.

**Convergence of the gradient flow.** In the statements of Theorems 3.3 and 3.5, the convergence of the Wasserstein gradient flow comes as an assumption. In order to prove convergence of gradient flows, one generally needs two properties: (i) compactness of the trajectories and (ii) a so-called *Łojasiewicz inequality* which, intuitively, controls how much a function flattens around its critical points. As compactness in $W_2$ is a very strong requirement, we have relaxed the topology where convergence is required to obtain more reasonable assumptions. Yet, even when a gradient flow lies in a compact set, there are some cases where it does not converge. There has been recent progress on related issues with the study of Łojasiewicz inequalities in Wasserstein space [5, 17], but to our knowledge, no general result is known in our non-geodesically convex case.

**Towards quantitative statements.** We stress that Propositions C.1 and C.4 provide with an intuitive criterion for a particle gradient flow to escape local minimum: roughly, it is sufficient that, when it passes close to a local minimum, at least one particle belongs to a $0$-sublevel set of the current potential $F'(\mu)$. In this paper we exploit this property by studying the many-particle limit, but other approaches are worth exploring. For instance, we could estimate the size of this sublevel set in specific cases, and use it as an indication for the particle-complexity to attain global minimizers. A discussion on a specific example is given in Section D.5.

# D  Case studies and numerical experiments

In this section, we verify the assumptions for the examples treated in Section 4.

## D.1  Loss functions

We first give sufficient conditions to satisfy the assumptions on the loss $R$, when the Hilbert space is $\mathcal{F} = L^2(\rho)$ for a probability measure $\rho$ on a space $\mathcal{X}$, which is either a domain of $\mathbb{R}^d$ or the torus. In this setting, typical losses are the form $R(f) = \int r(x, f(x)) \mathrm{d}\rho(x)$ for a function $r : \mathcal{X} \times \mathbb{R} \to \mathbb{R}_+$. The next lemma gathers some properties of such losses.

**Lemma D.1** (Properties of the loss)**.** *If $r$ is convex in the second variable, then $R$ is convex. If $r$ is differentiable in the second variable with $\partial_2 r$ Lipschitz, uniformly in the first variable, then $R$ is differentiable with differential $dR$ Lipschitz. If moreover $|\partial_2 r|^2 \leq C_1 r + C_2$ for some constants $C_1, C_2 > 0$, then $dR$ is bounded on sublevel sets.*

*Proof.* The convexity property is easy. If $\partial_2 r$ is $L$-Lipschitz (uniformly in the first variable), then it can be seen that $dR_f : h \mapsto \int r'(r, f(x)) h(x) \mathrm{d}\rho(x)$ is the differential of $R$ because for all $f, h \in \mathcal{F}$, by a Taylor expansion,

$$|R(f + h) - R(f) - dR_f(h)| \leq \frac{L}{2} \int |h(x)|^2 \mathrm{d}\rho(x) = \frac{L}{2} \|h\|^2 = o(\|h\|).$$

It is direct to see that $dR$ is $L$-Lipschitz in the operator norm. Finally, if $|\partial_2 r|^2 \leq C_1 r + C_2$, then

$$\|dR_f\|^2 = \int |\partial_2 r(x, f(x))|^2 \mathrm{d}\rho(x) \leq C_1 R(f) + C_2$$

so $dR$ is bounded on sublevel sets. $\qquad\qquad\qquad\qquad\qquad\qquad\qquad\qquad\qquad\qquad\square$

## D.2  Sparse deconvolution

Let us show that Assumptions 3.4 hold for the setting of Section 4.1. While we did not mention explicitly the choice of the torus $\Theta = \mathbb{R}^d / \mathbb{Z}^d$ as a domain in our results, it poses no difficulties: it is similar to the case $\Theta$ bounded, but without the difficulties related to boundaries. On the separable Hilbert space $\mathcal{F} = L^2(\Theta)$ where $\Theta$ is the $d$-torus endowed with the normalized Lebesgue measure, the loss $R$ is as in Lemma D.1 with $r(x, f) = (f(y) - y(x))^2$ and the regularization term $\tilde{V} = 1$

trivially satisfies the assumptions. Let us turn our attention to the function $\phi(\theta) : x \mapsto \psi(x - \theta)$. Its norm does not depend on $\theta$, so it is bounded. If $\psi$ is continuously differentiable with Lipschitz derivative, then $\phi$ is differentiable with $d\phi_\theta(\bar{\theta}) : x \mapsto \nabla\psi(x - \theta) \cdot \bar{\theta}$ which is bounded (again, its norm $\|d\Phi\| = \|\nabla\psi\|$ does not depend on $\theta$) and is Lipschitz, as similarly as in the proof of Lemma D.1.

It remains to check the Morse-type regularity assumption i.e., to check that for all $f \in \mathcal{F}$, the function $\theta \mapsto \langle f, \phi(\theta) \rangle = \int f(x)\psi(x - \theta)dx$ has a set of regular values which is dense in its range. If this function is constantly 0 then this is trivially true, otherwise, its range is an interval of $\mathbb{R}$. By Morse-Sard's lemma, if this function is $d - 1$-times continuously differentiable, then the set of critical values has zero Lebesgue measure and our assumption holds. By differentiating under the integral sign, this assumption is thus satisfied if $\varphi$ is $d - 1$-times continuously differentiable.

## D.3 Neural network: sigmoid activation

Let us show that Assumptions 3.4 hold for the setting presented in Section 4.2 in the case of sigmoid activation functions. We write the disintegration of $\rho$ with respect to the variable $x$ as $\rho(\mathrm{d}x \otimes \mathrm{d}y) = \rho(\mathrm{d}y|x) \otimes \rho_x(\mathrm{d}x)$ where $\rho_x$ is the marginal of $\rho$ on $\mathcal{X}$ and $(\rho(\cdot|x))_{x \in \mathcal{X}}$ a family of conditional probabilities on $\mathbb{R}$ (see [3, Thm. 5.3.1]). On the separable Hilbert space $L^2(\rho_x)$, the loss $R$ is as in Lemma D.1 with $r(x, p) = \int_{\mathbb{R}} \ell(p, y)\rho(\mathrm{d}y|x)$ and the regularization term $\tilde{V} = 1$ satisfies trivially the assumptions. In order to simplify notations, we consider the augmented variable $z = (x, 1) \in \mathbb{R}^{d-1}$ and $\rho_z$ its distribution when $x$ is distributed according to $\rho_x$. Let $\phi(\theta) : x \mapsto \sigma(z \cdot \theta)$, defined on $\Theta = \mathbb{R}^{d-1}$.

**Lemma D.2.** *If $\rho_x$ has finite moments up to order $4$, then the function $\phi : \mathbb{R}^{d-1} \to \mathcal{F}$ is differentiable with a Lipschitz and bounded differential $d\phi_\theta(h) : x \mapsto (h \cdot z)\sigma'(z \cdot \theta)$ where $z = (x, 1)$.*

*Proof.* Let us check that the function $d\phi$ defined above is indeed the differential of $\phi$. For $\theta, h \in \mathbb{R}^{d-1}$, we have

$$
\begin{aligned}
\Delta(h)^2 &:= \|\phi(\theta + h) - \phi(\theta) - d\phi_\theta(h)\|^2 \\
&= \int_{\mathcal{X}} |\sigma(\theta \cdot z + h \cdot z) - \sigma(\theta \cdot z) - (h \cdot z)\sigma'(z \cdot \theta)|^2 \mathrm{d}\rho_z(z) \\
&\leq \frac{L^2}{4} \int_{\mathcal{X}} |h \cdot z|^4 \mathrm{d}\rho_z(z)
\end{aligned}
$$

where $L$ denotes the Lipschitz constant of $\sigma'$. So if $\rho_z$ has finite 4-th order moment $M_4(\rho_z)$ then $\Delta(h) \leq \frac{L\sqrt{M_4(\rho_z)}}{2}|h|^2$ and $d\phi$ is indeed the differential of $\phi$. This differential is bounded and Lipschitz since $\|d\phi_\theta\| \leq \|\sigma'\|_\infty \sqrt{M_2(\rho_z)}$ and $\|d\phi_\theta - d\phi_{\tilde{\theta}}\| \leq L\sqrt{M_4(\rho_z)}|\theta - \tilde{\theta}|$ for all $\theta, \tilde{\theta} \in \mathbb{R}^{d-1}$. Finally, it is clear that if $\rho_x$ has finite 4-th moment then so does $\rho_z$. $\qquad\square$

It remains to check the Sard-type regularity assumption i.e., to check that for all $f \in \mathcal{F}$, $\theta \mapsto \langle f, \phi(\theta) \rangle = \int_{\mathcal{X}} f(x)\sigma((x, 1) \cdot \theta)\mathrm{d}\rho_x(x)$ has a set of regular values which is dense in its range. If this function is constantly 1 then this is trivially true, otherwise, its range is an interval of $\mathbb{R}$. If $\rho_x$ has finite moments up to order $2d - 2$ then the function above is $d - 1$ continuously differentiable and the conclusion follows by Morse-Sard's lemma.

In the statement of Proposition 4.2, the boundary assumption is explicitly mentioned so the proof is complete. We now briefly explain why is it difficult to check the Sard-type regularity in the boundary condition a priori. Consider the simple setting of a quadratic loss $R(f) = \frac{1}{2}\|f - f^*\|^2_{\mathcal{F}}$ where $f^*$ is the optimal Bayes regressor that we may assume smooth. As required in the boundary assumption, consider a function $f \in \mathcal{F}$ of the form $f = R'(\int \Phi \mathrm{d}\mu) = \int \Phi \mathrm{d}\mu - f^*$ for some $\mu$ in the domain of the functional $F$. In the limit $r \to \infty$, the function $g_f(r\theta) := \langle f, \phi(r\theta) \rangle = \int f(x)\sigma(r\theta \cdot (x, 1))d\rho_x(x)$ converges to the function $\bar{g}_f(\theta) = \int_{\theta \cdot (x, 1) \geq 0} f(x)d\rho_x(x)$. This function is continuously differentiable on the sphere if the density of $\rho_x$ is in $C_0(\mathbb{R}^{d-2})$ and $f$ is bounded and continuous (this is the case here) and the convergence of $g_f(r\cdot) \to \bar{g}_f$ is indeed in $C^1$. However, we cannot guarantee a very high regularity for $f$ in general: differentiating under the integral sign $d - 1$-times requires to have moments of order $(d - 1)$ bounded for $\mu$, which cannot be assumed a priori ($\mu$ is just known to be in the domain of $F$). This prevents us from applying Morse-Sard's lemma.

### D.4 Neural network: ReLU activation

#### D.4.1 Classical parameterization

We now consider the activation function $\sigma(s) = \max\{0, s\}$ and let $\Phi(w, \theta) : x \in \mathbb{R}^{d-2} \mapsto w\sigma((x, 1) \cdot \theta)$ be defined on $\mathbb{R} \times \mathbb{R}^{d-1}$. We show in the next lemma that $\Phi$ is not differentiable on the whole space: at points where the $\theta$ coordinate vanishes, there is a discontinuity in the derivative. The consequence of this Lemma is that particle gradient flows (Definition 2.2)—and a fortiori Wasserstein gradient flows—are not well-defined in this case.

**Lemma D.3.** *If $\rho_x$ has finite moments up to order 2 and has a density, then the function $\Phi : \mathbb{R}^d \to \mathcal{F}$ is differentiable on the set $\{(w, \theta) \in \mathbb{R} \times \mathbb{R}^{d-1} \; ; \; \theta \neq 0\}$, with differential $d\Phi_{(w,\theta)}(\bar{w}, \bar{\theta}) : x \mapsto (\bar{w} + w\bar{\theta} \cdot z)\sigma'(z \cdot \theta)$ where $z = (x, 1)$ and $\sigma'$ is the Heaviside step function. Yet, the differential $d\Phi$ is discontinuous at points of the form $(w, 0)$ for $w \neq 0$.*

*Proof.* Let us verify that the properties of a Fréchet differential are satisfied by the function $d\Phi$ above. For $u = (w, \theta)$ such that $\theta \neq 0$ and $\bar{u} = (\bar{w}, \bar{\theta})$ in $\mathbb{R}^d$, we have

$$\Delta_u^2(\bar{u}) := \|\Phi(u + \bar{u}) - \Phi(u) - d\Phi_u(\bar{u})\|^2$$
$$= \int |f(u + \bar{u}, x) - f(u, x) - df_{(u,x)}(\bar{u}, 0)|^2 \mathrm{d}\rho_x(x)$$

where we have introduced the function $f : (u, x) \mapsto w\sigma(\theta \cdot (x, 1))$ which is differentiable whenever $\theta \cdot (x, 1) \neq 0$. For $\theta \in \mathbb{R}^{d-1} \setminus \{0\}$ and $\epsilon > 0$, we introduce the sets $S_{\theta,\epsilon} = \{x \in \mathbb{R}^{d-2} \; ; \; |\theta \cdot (x, 1)| \leq \epsilon |(x, 1)|\}$ and decompose the previous integral in two parts: one where $f$ is regular and the integrand can be controlled with second order terms, and another one that deals with the non-differentiability inside $S_{\theta,\epsilon}$. This choice of definition for $S_{\theta,\epsilon}$ guarantees that we have $(\theta + \bar{\theta}) \cdot (x, 1) \neq 0$ whenever $x$ is not in $S_{\theta,\bar{\theta}}$. This leads to

$$\Delta_{(w,\theta)}^2(\bar{w}, \bar{\theta}) \leq \int_{S_{\theta,|\bar{\theta}|}} 4|(x, 1)|^2 \cdot |\bar{u}|^2 \cdot (2|u| + |\bar{u}|))^2 \mathrm{d}\rho_x(x) + \int_{\mathbb{R}^{d-2} \setminus S_{\theta,|\bar{\theta}|}} |\bar{w}\bar{\theta} \cdot (x, 1)|^2 \mathrm{d}\rho_x(x)$$
$$\leq 4|\bar{u}|^2 \cdot (2|u| + |\bar{u}|))^2 \int_{S_{\theta,|\bar{\theta}|}} |(x, 1)|^2 \mathrm{d}\rho_x(x) + \frac{1}{2}|\bar{u}|^4 \int_{\mathbb{R}^{d-2} \setminus S_{\theta,|\bar{\theta}|}} |(x, 1)|^2 \mathrm{d}\rho_x(x)$$

If $\rho_x$ has finite second order moment $M_2(\rho_x)$, then the second term is negligible in front of $|\bar{u}|^2$ when $|\bar{u}|$ goes to 0. In order to have the same property for the first term, it is sufficient that the integral $\int_{S_{\theta,|\bar{\theta}|}} |(x, 1)|^2 \mathrm{d}\rho_x(x)$ goes to 0 as $|\bar{\theta}|$ goes to 0 which is the case since $\rho_x$ has a density. Therefore, under these conditions, $\mathrm{d}\Phi_{(w,\theta)}$ is the differential of $\Phi$ at $(w, \theta)$. To exhibit a discontinuity, let $w \neq 0$ and $\theta \in \mathbb{S}^{d-1}$. For $t > 0$, it holds

$$\|d\Phi_{(w,t\theta)} - d\Phi_{(w,-t\theta)}\|^2 \geq |w|^2 \int |\theta \cdot (x, 1)|^2 \mathrm{d}\rho_x(x).$$

For suitable choices of $\theta$ (for instance, $\theta = (0_{\mathbb{R}^{d-1}}, 1)$), this lower bound is strictly positive and independent of $t$. $\qquad\square$

Although we do not use this fact explicitly in the paper, it is interesting to note that the regularizing potential $V : (w, \theta) \mapsto |w| \cdot |\theta|$ is admissible in the 2-homogeneous setting of Assumptions 3.2: although it is not differentiable nor convex, it is positively 2-homogeneous and semiconvex.

**Lemma D.4.** *The function $V : (w, \theta) \mapsto |w| \cdot |\theta|$ defined on $\mathbb{R} \times \mathbb{R}^{d-1}$ is positively 2-homogeneous and semi-convex.*

*Proof.* The homogeneity property is clear, and to see that $V$ is semi-convex, it is sufficient to remark that

$$(w, \theta) \mapsto V(w, \theta) + \frac{1}{2}|w|^2 + \frac{1}{2}|\theta|^2 = \frac{1}{2}(|\theta| + |w|)^2$$

is convex, since it is the square of a norm. $\qquad\square$

### D.4.2 A differentiable parameterization

We now consider the alternative parameterization considered in Proposition 4.3, defined as $\Phi(\theta) : x \mapsto \sigma(s(\theta) \cdot (x, 1))$ where $\sigma(t) = \max\{t, 0\}$ and $s$ is the signed square function $s(t) = t|t| = \mathrm{sign}(t) \cdot t^2$ that acts entry-wise. As $\Phi$ is clearly positively 2-homogeneous so we just have to prove the differentiability of $\Phi$, which is done with the same technique as in Lemma D.3.

**Lemma D.5.** *If $\rho_x$ has finite moments up to order 2 and has a density, then the function $\Phi : \mathbb{R}^d \to \mathcal{F}$ is differentiable, with differential $d\Phi_\theta(\bar{\theta}) : x \mapsto 2(\sum_{i=1}^{d} \bar{\theta}_i |\theta_i| z_i) \sigma'(s(\theta) \cdot (x, 1))$ where $\sigma'$ is the Heaviside step function.*

*Proof.* As in Lemma D.3, we verify that the properties of a Fréchet differential are satisfied by the function $d\Phi$ above. First, $\Phi$ is differentiable at 0 with differential 0 since it is positively 2-homogeneous. For $\theta \neq 0$ and $\bar{\theta}$ in $\mathbb{R}^d$, we have

$$\Delta_\theta^2(\bar{\theta}) := \|\Phi(\theta + \bar{\theta}) - \Phi(\theta) - d\Phi_\theta(\bar{\theta})\|^2$$
$$= \int |f(\theta + \bar{\theta}, x) - f(\theta, x) - df_{(\theta, x)}(\bar{\theta}, 0)|^2 \mathrm{d}\rho_x(x)$$

where we have introduced the function $f : (\theta, x) \mapsto \sigma(s(\theta) \cdot (x, 1))$ which is differentiable whenever $s(\theta) \cdot (x, 1) \neq 0$. For $\theta \in \mathbb{R}^d \setminus \{0\}$ and $\epsilon > 0$, we introduce the sets $S_{\theta, \epsilon} = \{x \in \mathbb{R}^{d-1} ; |s(\theta) \cdot (x, 1)| \leq \epsilon |(x, 1)|\}$ and decompose the previous integral in two parts: one where $f$ is regular and the integrand can be controlled with second order terms (through Taylor-Lagrange inequality), and another one that deals with the non-differentiability inside $S_{\theta, |\theta|}$ (where $f$ is only Lipschitz, locally in $\theta$ and globally in $(x, 1)$). This leads to the bounds, for some constants $C_\theta, C'_\theta > 0$ and $|\bar{\theta}|$ small enough

$$\Delta_\theta^2(\bar{\theta}) \leq C_\theta |\bar{\theta}|^2 \int_{S_{\theta, |\bar{\theta}|}} |(x, 1)|^2 \mathrm{d}\rho_x(x) + C'_\theta |\bar{\theta}|^4 \int_{\mathbb{R}^{d-1} \setminus S_{\theta, |\bar{\theta}|}} |(x, 1)|^2 \mathrm{d}\rho_x(x)$$

Under the assumption that $\rho_x$ has a density, we have that $\Delta_\theta^2(\bar{\theta}) = o(|\bar{\theta}|^2)$. Therefore, $\mathrm{d}\Phi_\theta$ is the differential of $\Phi$ at $\theta$. $\qquad\square$

Note that the condition on the moments of $\rho_x$ is less strong for ReLU activation than for sigmoids in Lemma D.2: this comes from the fact that ReLU is piece-wise linear. Similarly as what explained in the end of Section D.3, it is difficult to verify the Sard-type regularity assumption so it is left as an assumption in Proposition 4.3.

### D.5 Numerical experiments : details and additional results

**Animated particle gradient flows.** We display on Figure 6 animated versions[5] for each value of $m$ of the particle gradient flows shown in Section 4 as well as a particle gradient flow corresponding to the training of a neural network with a single hidden layer and sigmoid activation function in dimension $d = 2$. In all these cases, the global minimizer (supported on the dotted lines in the plots) is found.

**Setting for the empirical particle-complexity plot.** Here we give more details on the numerical experiments behind Figure 3.

1. For the leftmost panel, the setting is similar to that of Figure 1: for each realization, 5 spikes are randomly distributed on the 1-torus (with a minimum separation of 0.1) with random weights between 0.5 and 1.5 and a small noise is added to the filtered signal. Then for each choice of $m$, we initialize $m$ particles on a regular grid on $\{0\} \times \Theta$ and integrate the particle gradient flow with the forward-backward algorithm until the improvement per iteration is below a small tolerance threshold.

Figure 6: Particle gradient flows with a logarithmic time scale (animated). (left) sparse spikes deconvolution, with position $\theta$ shown horizontally and weight $w$ shown vertically. (center) neural network with ReLU activation function: we show for each particle the trajectory $|w(t)| \cdot \theta(t) \in \mathbb{R}^2$. (right) neural network with sigmoid activation function, with weights represented by the size of the particles (blue for negative, red for positive, ground truth shown with 2 neurons shown by large disks).

2. For the center panel, the setting is similar to that of Figure 2, but here in dimension $d = 100$. The data is normally distributed and the ground truth labels are generated by a similar neural network with 20 neurons (with random normally distributed parameters). The objective function is the square loss without regularization, so the global minimum corresponds to a 0 loss. We optimize using SGD with fresh samples at each iteration.

3. The rightmost panel shows, similarly, the particle-complexity for training a neural network with a single hidden layer and sigmoid activation function, in dimension $d = 100$. The data is distributed on a sphere and the ground truth labels are generated by a similar neural network with 20 neurons with random normal weights. Again, we minimize with SGD the square loss without regularization and the global minimum corresponds to a 0 loss.

We compare the performance with the method of simply minimizing on the weights with the same initialization. This is a convex problem, and the minimum value attained does not depend on the minimization method. We plot for each case the final excess loss as a function of $m$ for several random realizations of the experiment and, for each value of $m$, its geometric average over all realizations. We have indicated in transparent green the area of loss values which should be interpreted as "optimal" but are not exactly 0 because the optimization has been stopped in finite time and the loss is not known exactly but estimated through sampling.

**Choice of the initial weights in the partially $1$-homogeneous case.** In all previous numerical experiments dealing with the partially 1-homogeneous case, we have initialized the particle gradient flow on a discretization of $\{0\} \times \Theta$. But Theorem 3.5 allows for a large variety of initialization patterns. In this paragraph, we comment on the various possibilities and explain how the proof of Theorem 3.5 helps understanding why the corresponding particle-complexity is impacted.

We display on Figure 7 a sparse spikes deconvolution experiment, in a similar setting than in Figure 1, but with different initializations. For this problem, where $m_0 = 5$ spikes are to be recovered, we have observed numerically that the particle gradient flows initialized on a uniform grid on $\{0\} \times \Theta$ succeed in finding a global minimizer as soon as there are more than $m = 7$ particles. In the first panel of Figure 7, the particle gradient flow with $m = 15$ particles initialized on $\{1\} \times \Theta$ fails at finding a minimizer and a larger number of particles is needed for success (as shown in the center panel, with $m = 30$).

This phenomenon can be understood in light of the proof of Theorem 3.5: when $(\mu_t)_t$ enters the neighborhood (given by Proposition C.4) of the local minimum $\nu$ reached in the left panel, say at $t_0 > 0$, there exists a set $\mathbb{R}^- \times K^-$ such that if a particle of $\mu_t$ for $t > t_0$ falls in this set, then $(\mu_t)_t$ eventually escapes from this local minimum $\nu$. This set is, to put it simply, a 0-sublevel set of the function $F'(\nu)$, which is a positively 1-homogeneous function in the weight coordinate (the vertical axis in Figure 7). The difficulty here is that, because of the initialization, $\mu_{t_0}$ is concentrated on $\mathbb{R}_+ \times \Theta$, so we can only hope that a particle "slides" on a ridge of $F'(\nu)$ to eventually reach the set $\mathbb{R}^- \times K^-$. This is guaranteed to happen in the many-particle limit (this is the object of Lemma C.18), but this is likely to require a high density of particles around ridges of $F'(\nu)$ (the set

$\mathbb{R} \times \{\theta_0\}$ in the proof of Theorem 3.5). This supports the idea that initializing on $\{0\} \times \Theta$ is a good choice. In the rightmost panel of Figure 7 we also show the behavior for a uniform initialization on $(\{1\} \times \Theta) \cup (\{-1\} \times \Theta)$ which, in this example, also avoids the difficulty described above.

Figure 7: Particle gradient flow for partially 1-homogeneous problems (sparse spikes recovery): effect of the initialization pattern on the particle-complexity. (left) $m = 15$ particles on $\{1\} \times \Theta$: failure (center) $m = 30$ particles on $\{1\} \times \Theta$: success (right) $m = 10$ particles on $(\{1\} \times \Theta) \cup (\{-1\} \times \Theta)$: success.

## Footnotes

[5]This animation is not displayed correctly by certain document viewers. It is for instance compatible with Adobe Acrobat Reader.