[Reviews · NeurIPS 2018]

Reviewer 1



This paper studied the problem of minimizing a convex function of a measure. The authors focus on particle gradient descent algorithm, and showed in the limit of infinitely many particle and infinite small learning rate (thus follow gradient flow), the algorithm can converge to a measure which is the global minimizer of the problem. One layer neural network and sparse spikes deconvolution can be casted into the framework of this paper. I feel the title is slightly misleading (since the crucial property of global convergence relies on convex rather than over-parameterization). The theoretical results are also in some sense weak 1) it requires infinitely many particle which could be exponentially large; 2) it does not provide convergence rate, which might require exponentially many steps (thus intractable). Nevertheless, the idea and introduction of particle gradient decent and optimization on measure is quite interesting, and innovative. As one of the earliest work in this vein, I suggest the results may already worth acceptance to NIPS. Minor suggestion: I feel some efforts could be spent towards making the detail more transparent and easy to follow. In the part why training NN fall in the formulation of this paper, it might be easier to understand to directly say what is function R and G in those setting. Also the main paper seems never officially define what is 1-homogeneous or 2-homogeneous, although they are very important assumption for the theorems.

Reviewer 2



This paper considers the problem of optimizing over measures instead of parameters directly ( as is standard in ML), for differentiable predictors with convex loss. This is an infinite dimensional convex optimization problem. The paper considers instead optimizing with m particles (dirac deltas). As m tends to infinity this corresponds to optimizing over the measure space. Proposition 2.3 shows existence and uniqueness of the particle gradient flow for a given initialization. The main results show convergence of the particle gradient flow to the minimizer under infinite particle limit and under the assumptions of separability of the support of initialization. The paper has experiments on toy synthetic problems for sparse deconvolution and training one hidden layer neural nets, evaluating the performance of the finite particle method. The experiments show that for these toy problems, gradient flow with smaller number of particles (~100) converges to a global minimizer. Even though the theoretical results do not apply for ReLU networks, experiments show favorable performance for this class as well. Comments: The paper is a theoretical paper studying the optimization over measures using particle gradient flow, and provides conditions for convergence to global optima. It also has encouraging experiments on toy examples. The paper currently lacks discussion of limitations of this approach and what is required to scale it to solving reasonable ML tasks. The results also hold under some fairly strong assumptions on initialization and technical conditions on the functions. It will also be useful to discuss these assumptions a bit more on when do we expect them to hold in practice. In the introduction, it might be useful to introduce the optimization problem slowly with more details, as it is optimization on \mu and not \theta (as normally done). May be even highlight this difference to make it more clear. In line 25, the rectified linear unit doesn't fit the framework that requires \phi be differentiable. Subdifferential as defined in line 135 need not exist for a non-convex function right?. What happens to the subgradient flow then? Why is the condition on initialization required for proposition 2.5 but not for proposition 2.3? The description about gap between stationarity and optimality in lines 189-190 is not very clear. Other comments: R() is a poor choice of notation for the loss function, as it is confusing with the Real line and is also commonly used for the regularizer part. There is div() notation used in line 157 and referred to in appendix without defining it in the main paper, which reduces the readability of the paper. There has been some empirical work in neural nets that separately optimizes magnitude and direction of units of neural nets (https://arxiv.org/abs/1602.07868). It seems there is some high level similarity between these methods that might be interesting to explore more. Typos in lines 181, 220

Reviewer 3



This article studies the general problem of minimizing a loss functional on the space of measures. Specifically, the article provides sufficient conditions for the convergence of the Wasserstein gradient flow to converge to the global optimum. The results are derived under fairly general conditions which include specific applications of significant interest to the community. The paper is very well written--- the problem is motivated carefully, and the results are presented in a compelling fashion. The mathematical ideas used in the proofs are quite challenging, and might be useful in other related areas of research. I sincerely believe that the problem studied in the paper is of significant interest to this community. The authors support their results with numerical simulations. The statements of Theorem 3.3 and 3.5 assume convergence of the Wasserstein flows. The authors remark on the sufficient conditions for the aforementioned convergence in the supplementary material--- I suggest the authors move some/ all of this remark to the main article. It would be extremely insightful if the authors could at least provide guarantees for convergence of the Wasserstein flow in any specific application.